# Simulation of spatially distributed sources, transport, and transformation of nitrogen from fertilization and septic system in a suburban watershed

Ruoyu Zhang[1,2], Lawrence E Band[1,2], Peter M Groffman[3,4], Laurence Lin[1], Amanda K Suchy[3,5], Jonathan M Duncan[6], Arthur J Gold[7]

[1]Department of Environmental Sciences, University of Virginia, Charlottesville, 22904, USA
[2]Department of Civil and Environmental Engineering, University of Virginia, Charlottesville, 22904, USA
[3]Environmental Sciences Initiative, Advanced Science Research Center at The Graduate Center, City University of New York, New York, 10031, USA
[4]Cary Institute of Ecosystem Studies, Millbrook, 12545, USA
[5]Institute for Great Lakes Research and Biology Department, Central Michigan University, Mount Pleasant, 48858, USA
[6]Department of Ecosystem Science and Management, Pennsylvania State University, University Park, 16802, USA
[7]Department of Natural Resources Science, University of Rhode Island, Kingston, 02881, USA

*Correspondence to*: Ruoyu Zhang (rz3jr@virginia.edu)

**Abstract.** Excess export of reactive nitrogen in the form of nitrate ($NO_3^-$) from suburban watersheds is a major source of water quality degradation and threatens the health of downstream and coastal waterbodies. Ecosystem restoration and best management practices (BMPs) can be introduced to reduce in-stream $NO_3^-$ loads by promoting vegetation uptake and denitrification in the upland and riparian areas. However, accurately evaluating the effectiveness of these practices and setting regulations for nitrogen inputs requires an understanding of how human sources of nitrogen interact with ecohydrological systems. We evaluated how the spatial and temporal distribution of nitrogen sources interacts with ecohydrological transport and transformation processes along surface/subsurface flowpaths with respect to nitrogen cycling and export. Embedding distributed household sources of nitrogen and water within hillslope hydrologic systems influences the development of both planned and unplanned "hot spots" of nitrogen flux and retention in suburban ecosystems. We chose a well-monitored low-density suburban watershed, Baisman Run, in Baltimore County, Maryland, USA, to evaluate patterns of in-stream $NO_3^-$ concentrations and terrestrial nitrogen cycling processes in response to three common activities: irrigation, fertilization, and on-site sanitary wastewater disposal (septic systems). We augmented a distributed ecohydrological model, RHESSys, with estimates of the spatial distribution of these loads at household parcel level to develop a predictive understanding of the factors generating upland and riparian nitrogen cycling, transport, and stream $NO_3^-$ concentrations. We calibrate subsurface hydraulic parameters only without calibrating ecosystem and biogeochemical processes. The calibrated model predicted mean $NO_3^-$ concentrations of 1.43 mg $NO_3^-$-N L$^{-1}$ compared to the observed 1.6 mg $NO_3^-$-N L$^{-1}$ from water year 2013 to 2017. With spatially explicit irrigation, fertilizer, and septic effluent inputs, estimated denitrification rates in grass lawns, a dominant land cover in suburban landscapes, were also in the range of previously measured values. The highest predicted denitrification rates (N retention hot spots) were downslope of lawn and septic locations in a constructed wetland, and at a riparian sediment accumulation zone at the base of a gully receiving street drainage. These locations illustrate the development of hot spots for

nitrogen cycling and export in both planned and "accidental" retention features. Appropriate siting of suburban nutrient management and BMPs should assess and incorporate spontaneously developed nutrient hot spots to design improved landscape ecosystem N retention and water quality.

## 1    Introduction

Nitrogen (N) and carbon (C) are fundamental elements for ecosystem functions and are influenced by multiple factors including climate (Campo & Merino, 2016; Crowther et al., 2016), moisture and other soil properties (Pastor & Post, 1986; Wang et al., 2020), plant and microbial community composition (Chen et al., 2003), and human activities (Galloway et al., 2008). They are also influenced by the state and pattern of drainage flowpaths as different forms of C and N are mixed and transported to distinct edaphic conditions, potentially forming "hot spots" (McClain et al., 2003) that have a disproportionate influence on

landscape and watershed scale biogeochemical cycling functions. Understanding mechanisms of N and C cycling and interactions with hydrologic processes is necessary to design and implement efficient ecosystem service and restoration strategies. In urban, suburban, and exurban ecosystems, human disturbance to biogeochemical cycling has led to air and water quality degradation. Best management practices (BMPs) are popularly deployed to reverse the degradation and improve local and downstream water quality, increase C and N retention, and promote ecosystem resilience to prepare for extreme weather

events with changing climate. BMPs can be both structural (e.g., constructed wetlands) and non-structural (e.g., changing fertilization and irrigation regimes). In addition to planned BMPs, spontaneously developed "hot spots" (Palta et al., 2017) may be responsible for a large share of nutrient retention, and therefore should be identified and protected. Both planned and unplanned retention features exist at very localized, sub-hillslope scales. Therefore, gaining a comprehensive understanding of hillslope level ecohydrological behavior and interactions between i) ecosystems and human derived nitrogen sources, and

ii) flowpath modification, can lay the foundation for effectively mitigating these environmental issues through spatially well-conceived and sustainable management practices.

In urban ecosystems, human activities introduce additional inputs of water (e.g., lawn irrigation and septic effluent), carbon (e.g., mulch, lawn amendments) and nitrogen (e.g., septic systems, lawn and garden fertilization, sanitary sewer leakage), occurring on discrete land segments and altering watershed mass budgets of water and nutrients. Lawn fertilization can

contribute more than half of the total N input in urban watersheds, even if it is only applied to $20-30\%$ of the landscape (Band et al. 2005; Groffman et al., 2004; Hobbie et al. 2017).  In the United States, about 20% of households (26.1 million) are reported to be served by septic systems in 2007 (U.S. EPA, 2008). Through our work in the Baltimore Ecosystem Study, low density suburban areas have been shown to produce the highest $NO_3^-$ load per unit developed land among different land uses, degrading local and downstream water quality (Groffman et al., 2004; Zhang et al., 2022). Atmospheric deposition and septic

system wastewater N can comprise similar input amounts at the watershed scale, but septic input is concentrated over only 1-2% of the landscape, with a large, localized volume of wastewater sufficient to result in groundwater mounding and effluent plumes extending towards local streams (Cui et al., 2016). The concentrated inputs over limited areas by septic inputs and

lawn fertilization with or without irrigation creates delivery or retention patterns of N hot spots that provide opportunities for targeting N mitigation strategies (Groffman et al., 2023).

With rapid suburban and exurban sprawl, decision makers are facing environmental challenges which require detailed planning for siting BMPs effectively in watersheds to promote N retention, reduce N export in streams, and protect water quality. These include both constructed and "inadvertent" biogeochemical hot spots of N retention at specific hillslope locations (e.g., swales, wetlands, riparian areas) at resolutions required for landscape design. However, commonly used modelling frameworks often do not couple distributions and interactions of hillslope ecohydrological processes in transporting and transforming natural

and human-induced N sources to understand or predict local (neighbourhood or hillslope) scale transport and retention. Semi-distributed hydrologic models, such as the Storm Water Management Model (SWMM; Rossman, 2010) and the Soil Water Assessment Tool (SWAT; Arnold et al., 1998), are widely used to simulate nutrients loads at subwatershed levels outlets. They simulate water and nutrient balance based on Hydrologic Response Units (HRU) with similar land cover and soil, where nutrients are independently processed by BMPs and added to streamflow at the subcatchment outlet. However, these models

lack hillslope water and nutrient mixing along interacting surface/subsurface hydrologic flowpaths. These interactions are important to simulate the formation of biogeochemical hot spots where potential uptake and retention of nutrients are high. The lack of sub-hillslope flowpath processes may generate significant bias in estimating key hydrologic and biogeochemical processes (Band et al., 1993; Fan et al., 2019). Data-driven approaches, such as SPARROW (Ator & Garcia, 2016; Smith et al., 1997), are also developed to assess large scale water quality in streams by nonlinear regression from gauged discharge and

solute concentrations. However, these models also do not investigate hillslope-scale transport and transformation processes. In addition, there does not exist the data at hillslope scales to develop sufficient data-based approaches to understand and predict retention processes (e.g., denitrification, uptake, immobilization).

Fully distributed hydrology models, such as MIKE-SHE (Abbott et al., 1986a, 1986b) and RTM-PiHM (Bao et al., 2017; Zhi et al., 2022), ParFlow (Maxwell, 2013) and RHESSys (Regional Hydro-Ecological Simulator System, Tague & Band, 2004)

could explicitly couple hillslope hydrologic and biogeochemical processes that are required to understand transport and transformation of these human-induced N loads along hydrologic flowpaths from upland to stream. They simulate surface and subsurface hillslope processes with detailed topographic and soil information to generate distributed surface runoff, recharge, soil moisture, evapotranspiration (ET), and other ecohydrological variables. Lateral surface and subsurface drainages redistribute precipitation, resulting in gradients of water availability within a watershed from ridge to riparian areas. These

models include modules for biogeochemical reaction and transport processes, which can interact with the transport and storage patterns of soil water and provide high-resolution output for each location within a watershed.

Therefore, a spatially explicit and process-based framework that simulates hillslope hydrology and interactions between C, N, vegetation, water, and household-level human activities through flowpaths has important advantages to understand and manage non-point source pollutants and hot spots in urban watersheds (Bernhardt et al., 2017; Groffman et al., 2009). The ability to

represent processes at the scale of human perception can also provide information useful for decision making and community/stakeholder involvement. High-resolution simulations and visualization of spatially explicit water, N cycling, and

transport can facilitate understanding and communication of how human activities can alter terrestrial and aquatic ecosystem functions in urban ecosystems and contribute to participatory planning. The framework should be capable of extension to watersheds without water chemistry data which are less available than discharge records worldwide. It would be a valuable feature of the framework to estimate nutrient dynamics reasonably, while restricting calibration to hydrologic parameters. Calibrating nutrient dynamics may not allow generalization to watersheds without chemistry records or extrapolation to conditions in which water quality BMPs are implemented.

The RHESSys (Tague & Band, 2004) is an ecohydrological model that simulates mass balances of water, C, and N of a watershed including hydrologic and biogeochemical stores and cycling. The hydrologic component in RHESSys routes water and solutes based explicitly on topographic and infrastructure surface water flowpaths, and two-dimensional subsurface flow based on dynamic groundwater table gradients. Biogeochemical process rates are then estimated with modules modified from Biome-BGC (Running & Hunt, 1993), CENTURY$_{NGAS}$ (Parton et al., 1996) and subsequent models. In this study, we augmented RHESSys to include household-level transfer of groundwater for lawn irrigation and domestic water use, with domestic water use routed to septic spreading fields. By coupling hillslope hydrology and biogeochemistry at spatially connected patches, RHESSys estimates spatiotemporal patterns of soil moisture, lateral flow distribution, evapotranspiration, groundwater level, and N transportation, transformation, uptake, and immobilization. In summary, by adding modules of household-level lawn irrigation, fertilization, and septic releases (see Sect. 2.3), as commonly sourced from groundwater in low density suburban and rural areas, RHESSys is designed with the capacity to simulate the comprehensive ecosystem dynamics and feedbacks of introduced spatially explicit lawn irrigation, fertilization, and septic releases at resolutions commensurate with human management of the landscape. This facilitates the scientific assessment of small-scale human activity and modification to land cover and infrastructure in expanding suburban and exurban areas.

We developed and used the augmented version of RHESSys to investigate the spatial and temporal distribution of hydrologic and biogeochemical N cycling and export in a low-density suburban watershed, Baisman Run (BARN, see Sect. 2.1). BARN is in a suburban area of Baltimore County, with all households using septic systems and well water. We developed numerical experiments with and without human additions of water and N and compared model results to field observations for streamflow, water chemistry, and soil N cycling processes to answer the following research questions:

1) What are the individual and interacting contributions of different watershed N sources to streamwater N export?

2) How do the spatially nested patterns of water and N inputs from human activities alter spatial patterns of key ecohydrological processes including N retention, evapotranspiration, groundwater levels, and flows?

3) What are the patterns of hot spots for N retention and associated implications for the design of BMPs to promote N retention within suburban watersheds?

## 2 Methods

### 2.1 Study Area

Our study watershed (Fig. 1), BARN, is in Baltimore County, MD, outside of the urban sanitary sewer service boundary. The
3.8 km$^2$ watershed is in the Piedmont physiographic province with a rolling, locally steep landscape. Mean elevation is 170.5 m, with average slope 7.8°. Meteorological records from 1980 to 2018 were integrated from Baltimore/Washington International Airport (BWI) weather station and a local rain gage adjacent to BARN at the Oregon Park operated by the Baltimore Ecosystem Study (BES), available after 2013 (Welty & Lagrosa, 2020). Mean annual maximum and minimum

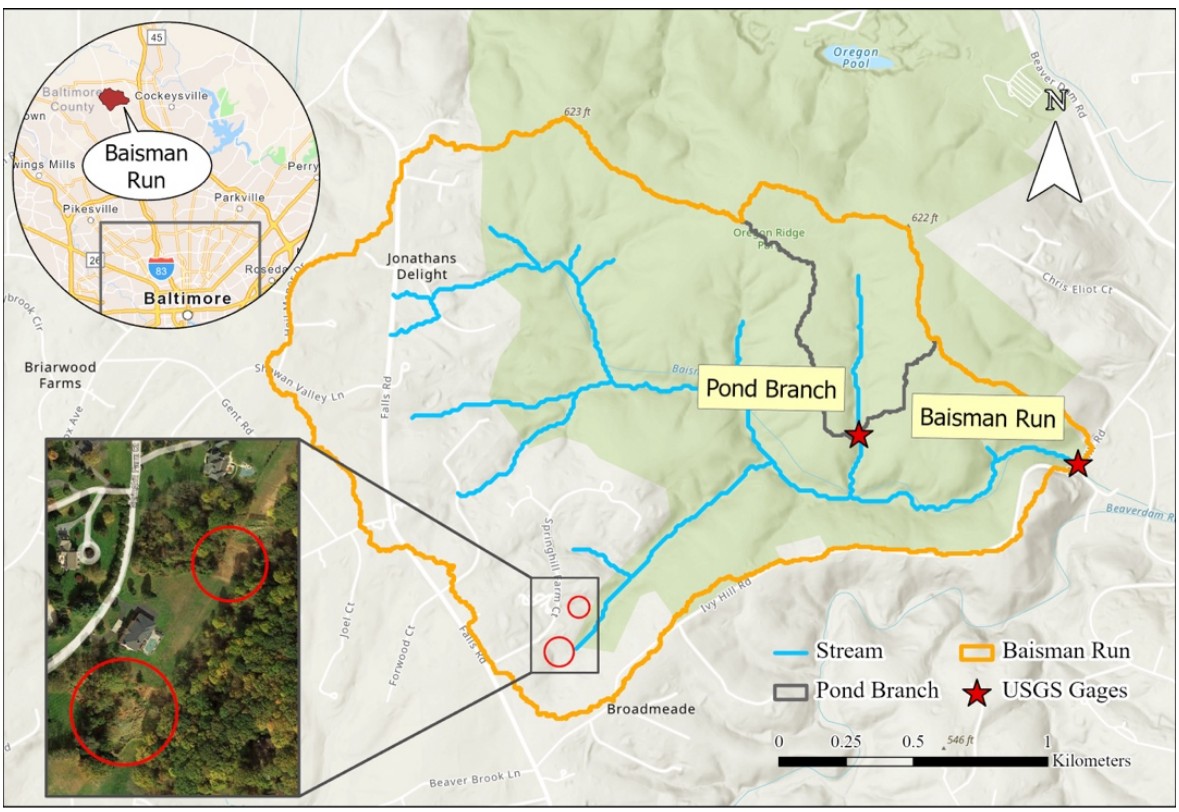

**Figure 1. Study watershed Baisman Run (BARN) in suburban Baltimore County, Maryland (from ESRI). The inset (from ESRI World Imagery) highlights two N retention "hot spots": A sediment accumulation zone (upper circle) receiving drainage from roads and a constructed wetland (lower circle). Pond Branch (POBR), a fully forested subcatchment is outlined in grey.**

temperatures are 18.9 °C and 7.9 °C respectively, and mean annual precipitation is 1,024 mm. The discharge and gage height records of BARN have been monitored by the USGS (Gage ID: 01583580) since 1999.

Soils in BARN range from silt clay loam to silt loam in the riparian areas to sandy loam on steeper slopes. Forested areas are dominated by approximately 100-year-old *Quercus spp.* (oaks) and *Carya spp.* (hickory). The entire watershed is underlain by the medium- to coarse-grained micaceous schist of the Loch Raven Formation, overlain by a weathered saprolite. The saprolite thickness is highest on ridges (up to 20m), thins (< 1 m) with some bedrock outcrops at steep midslope positions, and is 1–2 m in bottomland locations (Cleaves et al. 1970; St. Clair et al., 2015). Hydraulic conductivities of soils generally decrease with depth but may locally increase into the saprolite. The saprolite may store substantial amounts of moisture, and is drained through underlying bedrock fractures through a set of emergent springs on the valley sidewall-riparian area transition, providing a fairly steady baseflow (Putnam, 2018). Dominant land cover includes forest and lawns, covering 81.5% and 14.5% of the watershed, respectively. Impervious areas cover 4.0% of the watershed, including roofs of single-family houses, driveways and roads. Lawns are located in front and backyards of households in headwater areas of BARN. Two natural gas supply lines cut through the watershed, creating two strips of herbaceous land.

BARN is a useful watershed for examining the interactions between human activities and watershed ecohydrological response, as the sources and disposal of domestic water are on-site without external piped inputs and outputs. In this suburban watershed all households use groundwater wells for water supply and on-site septic systems to process wastewater. Lawn and garden fertilization is another major source of N input in BARN (Law et al., 2004). Septic and fertilization N and water additions are localized on lawns and septic drain fields near houses in the BARN headwaters.

The availability of several previously collected data sets allowed us to compare simulation results to field observations. Rich ecohydrological observations and lawn management surveys (Fraser et al., 2013; Law et al., 2004) from the BES are available as are weekly water chemistry concentration data at the BARN USGS gage since 1998 (Groffman et al., 2020; Castiblanco et al. 2023). In addition, a fully forested subcatchment of BARN, Pond Branch (POBR, Fig. 1), is also monitored weekly by the BES and USGS (Gauge ID: 01583570). POBR serves as a forest control site without human water and nutrient additions. Finally, we have previously estimated N stores and cycling rates, including lawn soil $NO_3^-$ content and denitrification rates in BARN (Suchy et al., 2023), sites on the campus of the University of Maryland Baltimore County (Raciti et al., 2011), and other sites in the region (Groffman et al., 2009). Annual atmospheric N deposition was estimated as 11 kg N ha$^{-1}$ from site MD99 of National Trends Network from National Atmospheric Deposition Program (NADP, 2022).

## 2.2  RHESSys setup and calibration

### 2.2.1  Model inputs and settings

Our study period makes use of observed and simulated watershed processes from water year 2013 to 2017 (i.e., Oct. 1, 2012 to Sep. 30 2017). BARN had gradual suburban development in the headwater which converted from agricultural land over a few decades. New development was largely completed in the 1990s, with one last field developed in 2007-2009. Our study period could reduce the uncertainty of N inputs due to land cover change during urban development and allow for analysis of N dynamics in a stationary condition. We set a 30-year simulation spin-up period to stabilize groundwater levels and C and N

pools. Inspection of the spin-up storage of soil C and N showed they were asymptotic with stable C:N ratios, with a mean of 8.5 in the entire watershed. The watershed is delineated using 1-m digital elevation data (Baltimore County GIS, 2017) using r.watershed (Ehlschleager et al., 2008) from GRASS GIS. Streams are identified when accumulated drainage areas are above 10 ha (Fig. 1), which approximates the extension of Baltimore County's hydrology lines dataset (Baltimore County GIS, 2016). Detailed land use information (Fig. A1) is derived from the 1-m high-resolution land use and land cover (LULC) data from the Chesapeake Conservancy (Claggett et al., 2018). The dataset contains "roof" as a LULC class, from which we identified 249 spatially isolated clusters of roofs within BARN. Comparison with the Baltimore County parcel dataset (Baltimore County GIS, 2019) and latest Google Earth satellite data allow us to filter out detached garages and sheds and to identify the main building in each parcel. We identified 181 households, although 13 homes are located on the watershed divide, providing some uncertainty to the effective number of septic systems.

We set up RHESSys in BARN at 10-m resolution. Patches in centroids of the 181 main buildings were identified as "drain-in" patches, receiving on site water supply from a hillslope scale groundwater store. We simulated groundwater well supplies from deep groundwater stores to household use as either domestic water use routed to septic spreading fields or to lawns for irrigation. Drain-in patches (homes) were paired with "drain-to" patches (septic spreading fields and lawns) to receive domestic water release, which are discussed in detail in Sect. 2.3. The methods can also draw water from ponds, but there is only one pond in the watershed that has occasionally been used for irrigation, and our simulations relied fully on groundwater wells. The riparian areas in RHESSys were defined as areas with height above nearest drainage (HAND) below 1.5 meters (Nobre et al., 2011). This is an approximation of riparian extent based on local inspection, and can be improved with more detailed riparian delineation. These areas were set to receive additional drainage from the deep groundwater system, which can set a feedback between greater household groundwater use and lower groundwater inputs to riparian areas. The start and end dates of the growing season in RHESSys are based on local observations and vary for lawn and forest: Deciduous tree grow from May 5[th] to Oct 22[nd]; Grass is set as perennial, identical to parameters in Lin et al. (2015 & 2019). There is limited conifer cover in BARN, and some Mountain Laurel (*Kalmia sp.*) understory on hillslopes .

### 2.2.2    Parameter calibration

We calibrate several subsurface hydraulic parameters to simulate lateral and vertical water flows and route subsurface lateral flow following the procedure detailed in Smith et al. (2022). In this study, we calibrated eight parameters (Table 1) for subsurface properties (i.e., lateral and vertical saturated hydraulic conductivities and their decay rates, pore size index, and air entry pressure) with initial estimates (Fig. A2) from the Soil Survey Geographic Database (SSURGO, USDA, 2019) and deeper groundwater processes (i.e., bypass seepage from surface and shallow saturated soil, and drainage rate to stream). We set the calibration period from water year 2013 to 2015 and validation period from water year 2016 to 2017. The original parameter values derived from SSURGO were further calibrated by multipliers to vary their magnitudes but preserve the spatial patterns of soil hydraulic properties (Fig. A2).

**Table 1. RHESSys parameters being calibrated and their physical representations (Tague and Band, 2004). Calibrated results shown as ranges of multipliers to original soil properties (Fig. A2 & A3) and groundwater component generating behavioral simulations with NSE greater than 0.5 for streamflow.**

| Parameter Groups | RHESSys Parameter Abbreviations | | Detail | Source | Unit | Multiplier Range |
|---|---|---|---|---|---|---|
| Lateral soil hydraulics | s | $m_l$ | Decay rate of lateral saturated hydraulic conductivity with depth | USDA SSURGO, 2019 | - | 0.31–2.91 |
| | | $K_{sat0\_l}$ | Lateral saturated hydraulic conductivity at the soil surface | | m day$^{-1}$ | 0.38–2.93 |
| | | $z$ | Soil depth | | m | 1.65–5.95 |
| Vertical soil hydraulics | sv | $m_v$ | Decay rate of vertical saturated hydraulic conductivity with depth | USDA SSURGO, 2019 | - | 0.51–1.98 |
| | | $K_{sat0\_v}$ | Vertical saturated hydraulic conductivity at the soil surface | | m day$^{-1}$ | 0.52–1.98 |
| Soil properties | svalt | $b$ | Pore size index | USDA SSURGO, 2019 | - | 0.51–1.98 |
| | | $\varphi_{ae}$ | Air entry pressure | | m | 0.5–1.05 |
| Groundwater dynamics | gw | $gw_1$ | Fraction of bypass from the saturated zone to groundwater storage | | - | 0–0.13 |
| | | $gw_2$ | Fraction of loss from groundwater storage to stream | | - | 0.03–0.5 |
| | | $gw_3$ | Fraction loss from surface to groundwater storage | | - | 0–0.07 |

Specifically, the simulated streamflow was used to calibrate against the daily USGS discharge records (Gage ID: 01583580). From four thousands of parameter set realizations randomly chosen within specified limits described in Smith et al. (2022), behavioral sets are chosen as yielding Nash-Sutcliffe efficiency (NSE; Nash & Sutcliffe, 1970) greater than 0.5 and fraction

of groundwater loss to stream (i.e., gw2 in Table 1) less than 0.5 to estimate the ensemble means and uncertainties of model simulations. The latter condition was enforced to regulate the flashiness of groundwater dynamics, as BARN is found to have large saprolite storage to provide steady baseflow (Putnam, 2018). To assess uncertainty, we reported the 95% uncertainty boundaries for simulated streamflow and $NO_3^-$ concentration and load. Lastly, we emphasized that no calibration was performed for N inputs (e.g., fertilization rate and septic load) or N cycling/transport processes in the model, as an important

aim of our methods is to evaluate the capacity of our model to regionalize to watersheds where no water chemistry but only streamflow observations were available.

### 2.3 Household additions of water and N

We included estimates of fertilization, onsite wastewater disposal from septic systems, and irrigation, as input to RHESSys to incorporate water and N management decisions and capture how such activities affect water and N cycling and export within the study watershed.

#### 2.3.1 Fertilization

The lawn fertilization module in RHESSys specifies the amount, location and timing of fertilization rates applied to lawns. Law et al. (2004) and Fraser et al. (2013) conducted in-person household surveys in a set of neighbourhoods in the Baltimore area, including BARN, and found that approximately 50% of homeowners apply fertilizer to their lawns, with a mean annual total fertilization rate ranging from 3.7 to 13.6 g N m$^{-2}$. Both surveys were conducted during significant drought conditions (2002 and 2008) when lawncare was reduced due to groundwater supply concerns. Hence, we consider the survey results to be on the lower end of actual rates. In this study, we used the intermediate lawn fertilization rates consistent with estimates of Law et al. (2004) surveyed in 2002, 8.4 g N m$^{-2}$ (12.4 kg N ha$^{-1}$ year$^{-1}$ at watershed scale, accounting for lawns that are not fertilized), for a denser suburban site. We assumed all lawns in BARN were fertilized three times with a 60-day interval between applications beginning April 1. This fertilization frequency is consistent with our prior household surveys and similar to results of surveys conducted in other suburban communities (Carrico et al., 2013; Martini et al., 2015). The model distributed the estimated total fertilization amount uniformly to all lawns in the watershed, at rates modulated by the proportion of lawns fertilized estimated by Law et al. (2004) and Fraser et al. (2013).

In the model, applied fertilizer is stored in an independent pool of each lawn patch, and each day we assumed a fixed fraction of available nutrients in the fertilizer pool leached to other pools, of which 80% is dissolved to detention storage and 20% to soil. Assuming all lawn fertilization is done with the slow-release fertilizer designed to remain 10% after one fertilization interval, the daily release fraction ($RF$) is determined by the fertilization interval ($FI$), following Eq. (1):

$$RF = -\frac{log\ 0.1}{FI}, \tag{1}$$

In our case study, our 60-day fertilization interval results in 3.8% of nutrients in the fertilization pool to decline exponentially and transported to other pools per day and then stored, consumed by vegetation, immobilized, denitrified or further transported to groundwater and downslope. User-defined fertilization time series could overwrite this setting of lawn fertilization if observations are available. In this study, we considered fertilizer input only contains NO$_3^-$, following sensitivity analysis that found varying NO$_3^-$ and NH$_4^+$ proportion in fertilizer had negligible impacts on model outputs. Once NO$_3^-$ is released to soil, N cycling is simulated following the procedure detailed in Lin et al. (2005). Phosphorous fertilizer, which is banned in Maryland lawn fertilizer formulations as protection for the Chesapeake Bay, is not considered.

## 2.3.2    Septic systems

All households within BARN use septic systems to dispose domestic wastewater. Wastewater from a house is released first to septic tanks for settling, then to drain fields which are typically placed downslope of the house. Therefore, soils in specified,

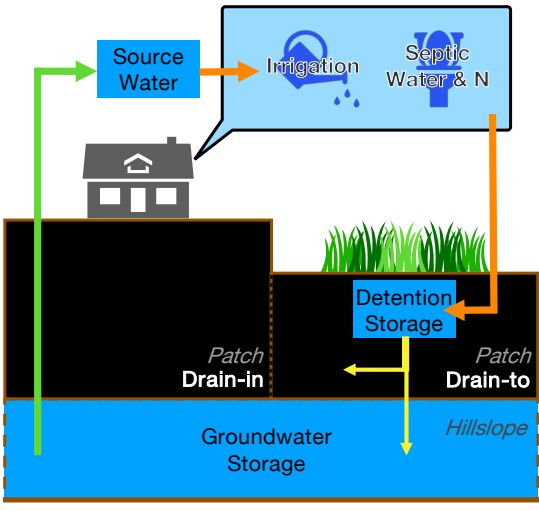

**Figure 2. Groundwater extraction for irrigation and septic systems in the RHESSys model. The source water (green arrow) is extracted from groundwater storage of drain-in patches (i.e., house centroids) and redistributed (orange arrow) to surface detention in downstream lawn patches for septic effluents and irrigated lawn patches of a household. After redistribution of source water, infiltration to soil and percolation to hillslope groundwater (yellow arrows) would follow the original processing of RHESSys**

downslope areas receive additional water and N input from septic effluents and may become N input hot spots in the watershed. We estimated the N load from septic systems as 7.7 kg N capita$^{-1}$ year$^{-1}$ and water input as 110.5 m$^3$ capita$^{-1}$ year$^{-1}$ (~80 gal$^{-1}$ capita$^{-1}$ day$^{-1}$), resulting in a NO$_3^-$ concentration of 70 mg N L$^{-1}$ estimated from results of Gold et al. (1990), Lowe et al. (2009), and other sources for per capita water use and septic nitrogen concentrations. We set the average number of people per
270 household as 3.3 for these single-family houses based on survey results from Law et al. (2004) and census information. Applying these water and NO$_3^-$ loads for 181 houses in BARN results in 4,599 kg N year$^{-1}$ (12.0 kg N ha$^{-1}$ year$^{-1}$) of NO$_3^-$ input to the watershed. The demand for septic source water ($SSW_{demand}$) is 66,001 m$^3$ year$^{-1}$ (17 mm year$^{-1}$ at the full watershed scale or 3,647 mm year$^{-1}$ normalized to the estimated total areas of septic fields) of water extracted from deep groundwater. Septic water and N loads are currently set to be evenly distributed through the year.
Septic source water is drawn from drain-in patches (i.e., centroid patches of main buildings) and transported to storage in septic drain-to patches (Fig. 2) which are the locations of drain fields of septic systems and defined as the closest downslope lawn patches to drain-in patches. We regulated actual withdrawal of septic source water ($SSW_{actual}$) to not exceed the available water in groundwater storage, as in Eq. (2):

$$SSW_{actual} = min(SSW_{demand}, GW_{storage}), \tag{2}$$

where $GW_{storage}$ is available water in surface detention and deep groundwater storage of the hillslope at drain-in patches (Fig. 2). This is a simplification which may miss more gradual household reduction of water use during droughts. The extracted source water is added to septic drain-to patches (orange arrow in Fig. 2), where it is subject to hydrological and biogeochemical processes. Nutrients are also added to the drain-to patches' storage, depending on concentrations and quantity of source water from the groundwater of drain-in patches. Once $NO_3^-$ is added to surface detention storage, N cycling is simulated following the procedure detailed in Lin et al. (2005).

### 2.3.3 Irrigation

Although irrigation practices and quantities vary significantly among households, irrigation is commonly applied during the growing season, and especially during dry and hot conditions. Therefore, we designed a mechanism to determine the total irrigation amount based on water stress of grass. Specifically, the amount of irrigation applied on lawns is determined by a water stress factor (*WSF*) in Eq. (3):

$$WSF = \frac{PET - ET}{PET}, \tag{3}$$

where PET and ET represent patch level potential and actual ET, which are estimated daily in RHESSys based on the Penman-Monteith equation (Monteith, 1965) and procedures in Sect. 5.6 in Tague & Band (2004). During continuously hot and dry days, WSF would increase due to lower soil water content (lower ET) and high atmospheric demand for water (higher PET). Our model then activates the irrigation function and calculates the demand of irrigation for patches modulated by water shortage. This function effectively modulates soil water conditions by the addition of groundwater sourced irrigation.

Unlike the septic source water ($SSW_{demand}$) which is fixed each day, the daily demand for irrigation source water ($ISW_{demand}$) in Eq. (4) for a lawn patch is further controlled by the water stress factor as:

$$ISW_{demand} = IR_{max} \cdot WSF \cdot lawn\%, \tag{4}$$

where $IR_{max}$ is the user-defined maximum daily irrigation rate, WSF is the water stress factor in Eq. (3), and lawn% is the fraction of grass in an irrigated patch. In the current model, we defined the maximum irrigation rate ($IR_{max}$) in BARN as 4 mm day$^{-1}$, which was converted based on the EPA's recommendation (U.S. EPA, 2024) of one inch per week for lawns. This rate can be modified based on the local practices or for sensitivity analysis. Like septic source water, withdrawal of irrigation source water cannot exceed available water in groundwater storage. The actual irrigation source water is calculated following the same rule in Eq. (1). The irrigation amount is pumped from deep groundwater storage to drain-in patches (i.e., centroids of houses, Fig. 2) to water lawns around houses. Irrigated lawns are limited to 50 m from houses, covering 33.7 ha (60.6%) out of 55.7 ha of lawns in BARN. We note that many households in this area are on programmed sprinkler systems, and our "smart" irrigation estimates may underestimate actual water use in non-drought conditions, but overestimate irrigation during droughts when homeowners reduce water use, contributing to input uncertainty. Dynamic water use is the subject of ongoing research in this watershed.

## 2.4 Scenarios and N hot spots

We focus on evaluating changes in $NO_3^-$ dynamics in riparian and upland areas when additional $NO_3^-$ is added from fertilization and/or septic systems, which resulted in four scenarios (Table 2) – *none* (no fertilization or septic inputs), *fertilizer only*, *septic only*, and *both* (fertilization and septic inputs) – for our study watershed. Irrigation is activated in all scenarios, including

our reference control scenario *none* to emphasize $NO_3^-$ dynamics without residential N inputs. Scenario *both* receives a total

**Table 2. Scenarios evaluated in BARN and corresponding combinations of augmented RHESSys features**

| Scenario Name | Irrigation | Fertilizer | Septic Processes |
|---|:---:|:---:|:---:|
| None | ✓ | | |
| Fertilizer Only | ✓ | ✓ | |
| Septic Only | ✓ | | ✓ |
| Both | ✓ | ✓ | ✓ |

of 35 kg N ha$^{-1}$ year$^{-1}$ of N input, with 11 (31.4%), 12 (34.3%), and 12 (34.3%) kg N ha$^{-1}$ year$^{-1}$ from atmospheric deposition, fertilization, and septic effluents, respectively, expressed at the watershed level. To better compare our $NO_3^-$ concentration results with the sampled weekly water chemistry from BES for BARN, we resampled the daily simulated concentration from RHESSys to weekly averages, expressed in unit of mg $NO_3^-$-N L$^{-1}$. The weekly $NO_3^-$ load was then estimated by the product of weekly mean $NO_3^-$ concentration and streamflow, expressed in unit of kg N ha$^{-1}$ year$^{-1}$. Note this approach may introduce bias for load as the once-a-week samples, typically not during major storms, and the observed daily mean discharges may not reflect the average load of the whole week.

We further evaluated changes in ecohydrological processes at potential on-site input hot spots (e.g., residential lawns and septic drainage fields) receiving direct household water and N inputs as well as off-site potential hot spots located in downslope areas that receive water and N inputs added upslope (e.g., riparian areas, wetlands, septic fields). Specifically, lawns are identified as patches with more than 50% of grass, and downstream forests are patches with more than 50% of forest downslope of residential area of BARN. One off-site location is a constructed wetland (lower red circle in Fig. 1), while the other is a spontaneously developed "accidental wetland" (Palta et al., 2017) in an area receiving road drainage and gully sedimentation, and is referred to as a "sedimentation accumulation zone" (lower red circle in Fig. 1).

In the Results section, we presented model calibration results in Sect. 3.1, in-stream $NO_3^-$ dynamics of scenarios in Sect. 3.2, and ecohydrological changes and N retention hot spots in Sect. 3.3, accordingly. Since no calibration was performed for N dynamics, $NO_3^-$ concentration and N retention processes were reported for the entire study period (i.e., water year 2013 to 2017).

## 3. Results

### 3.1 Model calibration and validation on streamflow

After performing calibration on soil hydraulic and groundwater parameters, we found 50 behavioral parameter sets that provided simulations meeting the requirement in Sect. 2.2.2. The range of calibrated multipliers are listed in Table 1, and the distributions are shown in Fig. A3. In the calibration period (i.e., water year 2013 to 2015, Fig. 3a), the ensemble of simulated

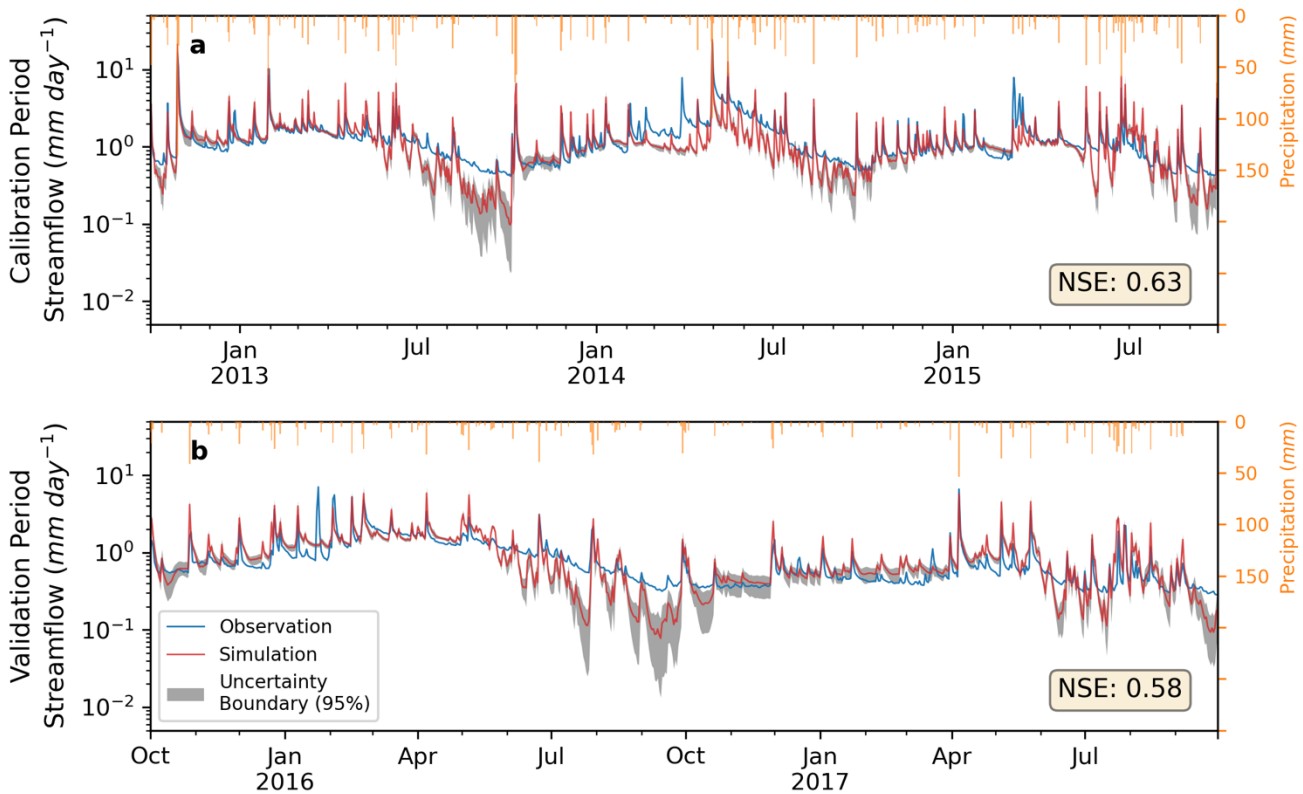

**Figure 3. The ensemble mean of daily streamflow from simulations (red), USGS observations (blue), and the daily 95% uncertainty range (grey) from 50 simulations with NSE greater than 0.5. The periods of (a) calibration was from Oct. 2012 to Sep. 2015 and (b) validation from Oct. 2015 to Sep. 2017. All simulations include irrigation, lawn fertilization, and septic processes**

mean (standard deviation) daily streamflow was 1.24 (±0.03) mm day$^{-1}$, with NSE of 0.63 (between 0.5 and 0.69) compared to the USGS observed 1.38 mm day$^{-1}$. In the validation period (Fig. 3b), the simulated ensemble mean (standard deviation) streamflow was 0.91 (±0.03) mm day$^{-1}$, with NSE of 0.58 (between 0.44 to 0.64) compared to the USGS's 0.86 mm day$^{-1}$. Negligible difference was detected after activating lawn fertilization or septic processes in the watershed. The small drop of NSE in the validation period compared to the calibration period indicated that our calibrated parameters reasonably captured

the hydrologic behavior of BARN.  The 95% uncertainty boundary encompassed the majority of observed moderate flows. Our model was able to simulate the seasonality of streamflow and water balance well compared to the observation records, but tended to underestimate the lowest flows in the growing season (from May to September) when streamflow is lowest and dominated by baseflow. During the entire study period, the mean simulated growing season streamflow was 0.95 mm per day which is 0.13 mm (-12%) per day lower than the 1.08 mm per day in the USGS records.

**Table 3. Mean weekly $NO_3^-$ concentration (mg N $L^{-1}$) and load (kg N $ha^{-1}$ $year^{-1}$) from calibrated simulations for BES weekly observations (BARN and POBR) and RHESSys simulation scenarios in each season and the entire study period from water year 2013 to 2017. Standard deviations from behavioral simulations for all scenarios were included below the mean values.**

| Variables | Season | Observation | | RHESSys Scenarios | | | |
| --- | --- | --- | --- | --- | --- | --- | --- |
| | | BARN | POBR | Both | Septic Only | Fertilizer Only | None |
| Concentration (mg N $L^{-1}$) | Spring | 1.5 | 0.02 | 1.4 (±0.12) | 0.76 (±0.08) | 0.77 (±0.05) | 0.27 (±0.03) |
| | Summer | 1.6 | 0.07 | 1.26 (±0.13) | 0.68 (±0.1) | 0.79 (±0.1) | 0.33 (±0.06) |
| | Fall | 1.57 | 0.06 | 1.41 (±0.23) | 0.77 (±0.15) | 0.94 (±0.17) | 0.41 (±0.09) |
| | Winter | 1.75 | 0.01 | 1.63 (±0.18) | 0.88 (±0.12) | 0.96 (±0.1) | 0.35 (±0.05) |
| | Mean | 1.6 | 0.04 | 1.43 (±0.16) | 0.77 (±0.11) | 0.87 (±0.1) | 0.34 (±0.06) |
| Load (kg $ha^{-1}$ $year^{-1}$) | Spring | 10.93 | 0.01 | 8.86 (±0.63) | 4.84 (±0.42) | 4.77 (±0.31) | 1.62 (±0.16) |
| | Summer | 5.88 | 0.02 | 4.72 (±0.36) | 2.49 (±0.25) | 2.81 (±0.23) | 1.06 (±0.16) |
| | Fall | 4.72 | 0.01 | 4.72 (±0.39) | 2.57 (±0.26) | 3 (±0.27) | 1.23 (±0.16) |
| | Winter | 8.38 | 0.01 | 8.42 (±0.68) | 4.61 (±0.46) | 4.91 (±0.38) | 1.81 (±0.18) |
| | Mean | 7.44 | 0.01 | 6.68 (±0.47) | 3.63 (±0.33) | 3.87 (±0.27) | 1.44 (±0.16) |

## 3.2 Improved prediction of $NO_3^-$ export

Turning fertilization and septic processes on and off in the model produced variation in in-stream $NO_3^-$ concentration and load simulations. We calculated weekly means of $NO_3^-$ load and concentration of behavioral simulations. In our 5-year study period, the ensemble mean $NO_3^-$ concentrations (Fig. 4a) for scenarios *none*, *septic only, fertilizer only, and both* were 0.34, 0.77,

0.87, and 1.43 mg NO$_3^-$-N L$^{-1}$, respectively (Table 4). The mean long-term observed concentration at the BARN USGS gauge was 1.6 mg NO$_3^-$-N L$^{-1}$. Thus, the simulated bias of mean NO$_3^-$ concentration considering both fertilization and septic loads decreased significantly from -1.26 mg NO$_3^-$-N L$^{-1}$ in the scenario *none* to 0.17 mg NO$_3^-$-N L$^{-1}$ in the scenario *both*. The 95%

uncertainty boundary of weekly NO$_3^-$ concentration in scenario *both* captured 67% of the weekly sampled observations. The seasonality of NO$_3^-$ concentration is also well captured, except for the growing season (e.g., Jul. to Oct. in 2013 and 2016) when the model underestimated low flows (Sect. 3.1). At seasonal scales (Table 3), the weekly mean NO$_3^-$ concentrations of scenario *both* from spring to winter were underestimated compared to the BES observations by small amounts (0.1 (-7%),

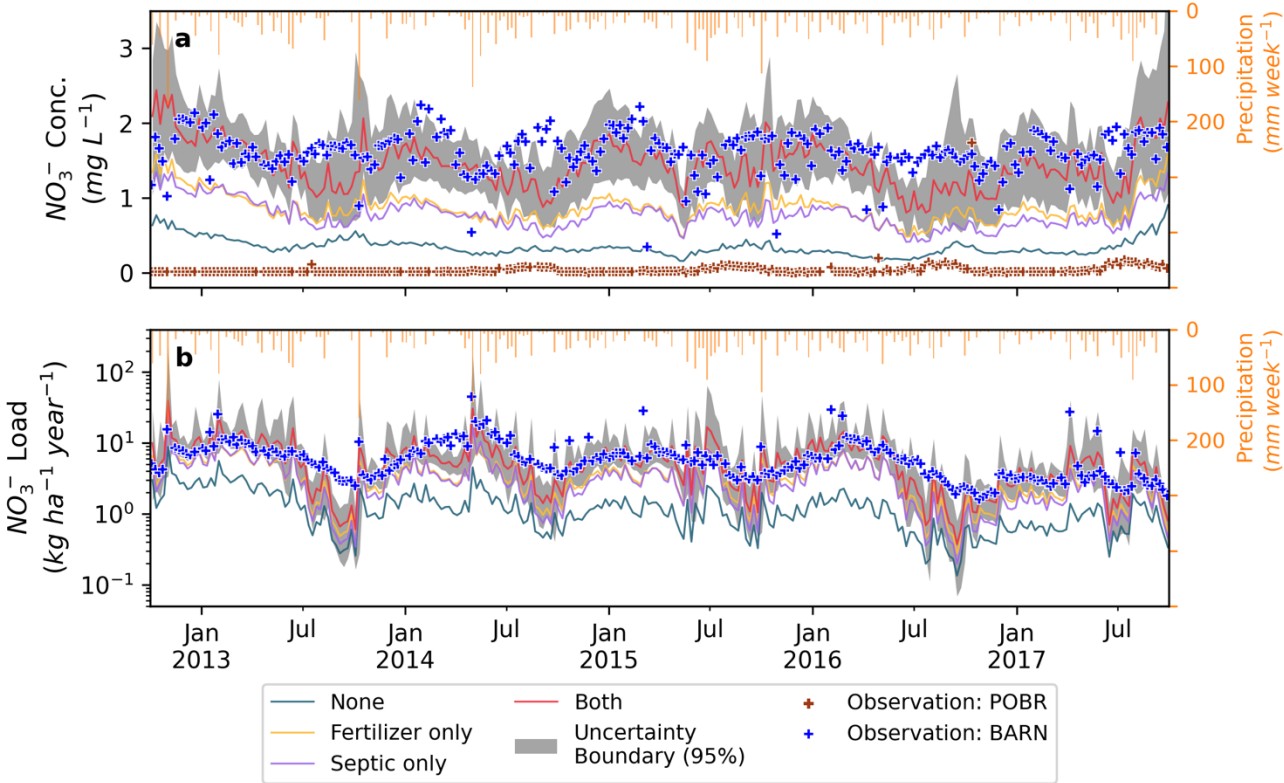

**Figure 4. Ensemble weekly mean of (a) NO$_3^-$ concentration and (b) load at the outlet of Baisman Run over the entire study period (water year 2013 to 2017). The 95% uncertainty boundary for scenario *both* is shown in grey.**

0.34 (-21%), 0.16 (-7%), and 0.12 (-10%) mg NO$_3^-$-N L$^{-1}$). The highest underestimation of NO$_3^-$ concentration in summer was aligned with the period that our model underestimated the lowest flows in the growing season (Sect. 3.1).

The in-stream $NO_3^-$ load (Fig. 4b) followed a similar trend as concentration, and the bias was reduced substantially from scenario *none* to *both* when fertilizer and septic loads were included. Scenario *none* underestimated $NO_3^-$ load by 6 (-81%) kg $NO_3^-$-N ha$^{-1}$ year$^{-1}$, and the scenario *both* decreased the bias substantially to -0.77 (-10%) kg $NO_3^-$-N ha$^{-1}$ year$^{-1}$. The seasonality was also well simulated by our model. The ensemble mean loads (Table 3) in fall and winter were accurately captured with close-to-zero bias compared to the observations, and the bias in spring and summer was slightly higher. The differences were due to lower simulated than observed discharges (Fig. 3) during the growing season. Lastly, the $NO_3^-$ retention rate (i.e., % of N input not exported in streamflow) varied across different scenarios ranging from a high of 87% in scenario *none* (atmospheric deposition only) to a low of 81% in scenario *both*. In scenarios *septic only* and *fertilizer only*, retention rates were 84% and 83%, respectively.

### 3.3 Ecohydrological and biogeochemical responses of hot spots

In our simulations, fertilizer is slowly released to soil and surface detention and transported downslope. This transport is augmented by irrigation and septic inputs. As a result, water and $NO_3^-$ are redistributed through other patches along subsurface hydrological flowpaths, providing "off-site" ecohydrological and biogeochemical responses downslope and across the whole watershed.

#### 3.3.1   Soil moisture and ET

The ensemble catchment mean of water table depth (Fig. A4) from all behavioral simulations under scenario *none* was 4.52 m during the study period. Fertilization had overall negligible effects on watershed mean soil moisture or water table depth compared to the base (*none*) scenario (Fig. 5a–5c), but minor increase of water table depth was detected in the residential areas, likely due to higher ET in lawns after fertilization. Septic processes decreased mean catchment water table depth to 4.47 m by groundwater mounding, which increases shallow groundwater flow to surrounding patches along connected flowpaths. Specifically in septic drainage field patches, the mean water table depth decreased to 3.69 m (-0.66 m, -15%) in scenarios *both* and 3.72 m (-0.63 m, -14%) in *septic only* compared to the mean depth of 4.35 m, in scenarios *none* and *fertilizer only*. With hillslope groundwater as the only source for septic process, we found groundwater withdrawal resulted in slightly drier conditions (i.e., increase of water table depth) in riparian areas of these residential hillslopes (Fig. A7, hillslopes 11 to 16), where the mean water table depth increased by 5 mm (2%) and 8 mm (3.4%) in scenarios *septic only* and *both* compared to 219 mm depth in scenarios *none* and *fertilizer only*. Though the standard deviation of each scenario from the 50 behavioral simulations was 1.1 m, the spatial distribution of soil moisture is consistent among all behavioral simulations.

The watershed-scale mean ET was 43.9 mm month$^{-1}$ in scenario *none* and 44.0 mm month$^{-1}$ in scenario *fertilizer only*. The standard deviation from 50 behavioral parameter sets was 0.8 mm month$^{-1}$ for each scenario. As the result of higher soil moisture levels after activating septic processes in scenario *both*, ET in lawn patches and septic drainage fields increased to (by) 42.3 (+0.4 mm month$^{-1}$, 1.0%) and 40.8 (+6.5 mm month$^{-1}$, 18.9%) mm month$^{-1}$, compared to the levels in scenarios *none*, respectively. With septic processes activated, mean ET increased to 44.1 and 44.2 mm month$^{-1}$ in scenarios *septic only* and

*both* in the residential hillslopes, which could be contributed by the additional water extracted from groundwater to surface

soil at the upland areas (in Fig. 5). When fertilization is activated in scenario *fertilizer only*, ET in riparian areas of residential hillslopes decreased to (by) 54.7 (-0.1 mm month[-1], -0.3%) mm month[-1] compared to scenario *none*, while the upland of these hillslopes increased by 0.1 mm month[-1]. This showed that fertilization in the upland residential lawns could support higher growth rate of vegetation but reduced water from draining towards downstream areas of a hillslope (in Fig. 5).

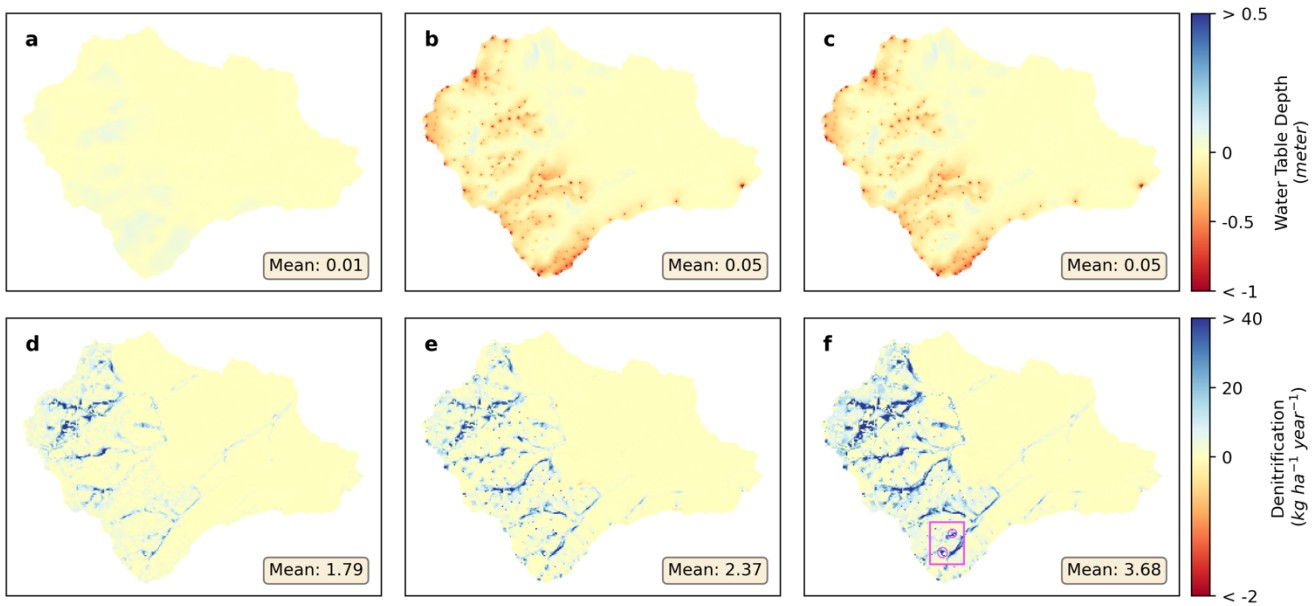


**Figure 5. Ensemble mean differences of water table depth (top panel) and denitrification (lower panel) between scenario *none* and scenario *fertilizer only* (a & d), *septic only* (b & e), and *both* (c & f). The inset highlights two hot spots of denitrification (i.e., wetlands in Fig. 1) were circled in (f).**

**3.3.2    Denitrification**

Our model suggested significant changes in denitrification after including additional $NO_3^-$ inputs from fertilization and septic processes. Compared to scenario *none* (Fig. A5), the ensemble mean annual rates of denitrification at the watershed scale were 7.2, 7.8, and 9.1 kg N ha[-1] year[-1] in scenarios *fertilizer only*, *septic only*, and *both*, respectively, increasing by 33%, 44%, and 68% (Fig. 5d–5f & Table 4). The standard deviation from the 50 behavioral simulations was 1.5 kg N ha[-1] year[-1] for scenario

*none* and *fertilizer only* and 1.6 kg N ha[-1] year[-1] for scenario *septic only* and *both*. When fertilization and septic processes were activated, the denitrification rates increased at the residential hillslopes and their riparian areas. The only exception was found in scenario *septic only*, where 7 patches experiencing minor reduced denitrification (-1.4% in average). All these patches were

found in riparian areas of residential hillslopes where the water table drops by 9 mm in average after the septic processes extracting groundwater in the upstream.

Compared to scenario *none*, denitrification rates increased significantly in hot spots – lawn, septic drainage field, and riparian areas (Table 4) in response to $NO_3^-$ inputs from fertilization and septic processes. Scenario *fertilizer* had higher denitrification rates than scenario *septic only* in lawns. Denitrification rates in septic drainage patches was increased by 368% in scenarios *septic only* compared to in the reference scenario *none* where these patches do not receive additional water and N inputs. Fertilization and septic processes added more than 20 kg N ha$^{-1}$ year$^{-1}$ load at the watershed level concentrated in upland

residential areas. These additions increased mean denitrification rates in forest patches in and below residential areas (i.e., excluding patches in Pond Branch) by 72.7% (Table 4). The annual denitrification rates in the sediment accumulation zone (upper red circle in Fig. 5) showed a significant increase after activating fertilization and septic processes, from 76.9 kg N ha$^{-1}$ year$^{-1}$ before to 95.6 (+18.7, 24.3%) kg N ha$^{-1}$ year$^{-1}$ after activation. Similarly, denitrification rates in the constructed wetland (lower red circle in Fig. 1) increased from 81.5 kg N ha$^{-1}$ year$^{-1}$ before to 102.7 (+21.2, 26%) kg N ha$^{-1}$ year$^{-1}$ after activation.

Changes in denitrification varied among seasons (Table 4). At the watershed scale and in all hot spots, the highest rates were generally found in spring and summer, followed by fall, and lowest in winter. The greatest increases (%) in denitrification at all locations were in spring when fertilizer is applied to lawns and soil moisture is generally higher. Riparian areas had significant increases in denitrification in winter when the watershed receives sustained $NO_3^-$ input from septic effluents.

Our modelled denitrification rates are consistent with measurements from field studies in Baltimore. Assuming 210 days (~7

months) that denitrification would occur, Raciti et al. (2011) reported a denitrification rate of 204 kg N ha$^{-1}$ year$^{-1}$ at 20 °C for saturated soil samples from fertilized lawns at the University of Maryland Baltimore County. At the same temperature, Suchy et al. (2023) reported a higher rate, 744 kg N ha$^{-1}$ year$^{-1}$, when lawn soil samples collected from BARN lawns were saturated. We interpolated the two rates based on the method from Raciti et al. (2011), assuming 5% storm (i.e., saturated soil) and 95% dry (i.e., low-soil-moisture) days (with a denitrification rate of 2.95 kg N ha$^{-1}$ year$^{-1}$) rate in a year. The projected climate-

adjusted mean denitrification rates were 13 and 40 kg N ha$^{-1}$ year$^{-1}$ from Raciti et al. and Suchy et al, which are very similar to estimates of annual denitrification from our simulated scenarios (Table 4). The mean 25 and 85 percentiles of annual denitrification rate for lawns from all simulations in scenario *both* were 2.8 to 30.8 kg N ha$^{-1}$ year$^{-1}$, respectively, which are comparable with the range of empirical measurements from low to high soil moisture conditions and various fertilization rates.

## 4  Discussion and Conclusions

### 4.1  Hydrologic processes

In BARN, household water use from wells transports roughly 0.05 mm day$^{-1}$ of water from groundwater to septic systems at the watershed level (10 mm day$^{-1}$ on septic fields). However, the conversion of groundwater to septic usage produced only negligible changes in streamflow, while locally changing soil moisture and groundwater levels. Specifically, simulated

streamflow was slightly decreased compared to the condition without septic water input. Inspecting growing season phenology,

we found both ET and net photosynthesis were elevated with septic input (Fig. A8). This may be due to local increases in septic water and nutrients increasing ET during the growing season, reducing groundwater recharge, in addition to the reduced groundwater storage, and contribution to watershed baseflow. We also noted that our model tended to underestimate the lowest streamflows during the growing season, which was also found in another suburban watershed, Dead Run, in Baltimore by Miles (2014). Several potential reasons could cause this discrepancy: 1) Higher transpiration estimates caused by uncertainties


**Table 4. Ensemble mean denitrification rates (kg N ha⁻¹ year⁻¹) in different locations under four scenarios and all seasons. Absolute and relative changes between scenario *none* and other scenarios are included below denitrification rates. Rates for forest excluded Pond Branch patches where no fertilizer or septic inputs are added.**

| Location | Season | Scenario | | | |
|---|---|---|---|---|---|
| | | None | Fertilizer Only | Septic Only | Both |
| Lawn | Spring | 9.4 | 13.3 (3.9, 41.8%) | 11.7 (2.3, 24.7%) | 15.0 (5.6, 59.4%) |
| | Summer | 11.6 | 16.0 (4.4, 38%) | 13.9 (2.3, 19.8%) | 17.6 (6, 51.6%) |
| | Fall | 9 | 11.3 (2.3, 25.7%) | 10.9 (1.9, 20.9%) | 12.7 (3.7, 41.4%) |
| | Winter | 6.6 | 8.1 (1.5, 22.7%) | 8.3 (1.7, 26.4%) | 9.4 (2.8, 42.4%) |
| | Annual | 9.2 | 12.3 (3.1, 33.4%) | 11.3 (2.1, 22.4%) | 13.8 (4.6, 49.7%) |
| Septic Fields | Spring | 3.8 | 5.8 (2, 52.4%) | 18.7 (14.9, 392.6%) | 18.7 (14.9, 391.6%) |
| | Summer | 4.7 | 6.1 (1.4, 30.6%) | 19.8 (15.1, 321.1%) | 19.7 (15, 319.4%) |
| | Fall | 4 | 4.8 (0.8, 19.5%) | 19.4 (15.4, 385.8%) | 19.3 (15.3, 383%) |
| | Winter | 3.2 | 4.0 (0.8, 24.7%) | 15.0 (11.8, 368.4%) | 15.0 (11.8, 367.8%) |
| | Annual | 3.9 | 5.2 (1.3, 33.3%) | 18.3 (14.4, 367.9%) | 18.2 (14.3, 366.2%) |

| | | | | | |
|---|---|---|---|---|---|
| **Riparian Areas** | **Spring** | 13.2 | 20.9<br>(7.7, 58.3%) | 23.4<br>(10.2, 76.9%) | 28.3<br>(15.1, 114.2%) |
| | **Summer** | 14.5 | 19.1<br>(4.6, 31.9%) | 19.9<br>(5.4, 37.2%) | 23.8<br>(9.3, 63.9%) |
| | **Fall** | 11.1 | 15.7<br>(4.6, 41.7%) | 16.0<br>(4.9, 43.9%) | 19.7<br>(8.6, 77.3%) |
| | **Winter** | 10.1 | 15.4<br>(5.3, 52.8%) | 16.7<br>(6.6, 65.1%) | 19.8<br>(9.7, 95.5%) |
| | **Annual** | 12.3 | 17.9<br>(5.6, 45.4%) | 19.1<br>(6.8, 55.3%) | 23.0<br>(10.7, 87.1%) |
| **Forest** | **Spring** | 6.5 | 8.7<br>(2.2, 34.5%) | 10.4<br>(3.9, 59.7%) | 11.7<br>(5.2, 80.2%) |
| | **Summer** | 3.8 | 5.0<br>(1.2, 30.8%) | 5.5<br>(1.7, 45.5%) | 6.7<br>(2.9, 75%) |
| | **Fall** | 3.8 | 5.0<br>(1.2, 30.8%) | 5.4<br>(1.6, 41.8%) | 6.4<br>(2.6, 68.4%) |
| | **Winter** | 4.9 | 6.3<br>(1.4, 29.4%) | 7.4<br>(2.5, 50.8%) | 8.2<br>(3.3, 67.1%) |
| | **Annual** | 4.8 | 6.3<br>(1.5, 31%) | 7.2<br>(2.4, 50.4%) | 8.3<br>(3.5, 72.7%) |
| **Watershed** | **Spring** | 7 | 9.5<br>(2.5, 35.3%) | 10.4<br>(3.4, 48.1%) | 12.1<br>(5.1, 72.4%) |
| | **Summer** | 5.1 | 6.9<br>(1.8, 34.3%) | 7.0<br>(1.9, 36.3%) | 8.5<br>(3.4, 65.9%) |
| | **Fall** | 4.7 | 6.1<br>(1.4, 29.4%) | 6.4<br>(1.7, 35.7%) | 7.5<br>(2.8, 59.1%) |
| | **Winter** | 5.2 | 6.6<br>(1.4, 26%) | 7.4<br>(2.2, 42.7%) | 8.2<br>(3, 58.5%) |
| | **Annual** | 5.4 | 7.2<br>(1.8, 32.9%) | 7.8<br>(2.4, 43.6%) | 9.1<br>(3.7, 67.6%) |


in vegetation ecophysiological parameters in RHESSys controlling vegetation water use or phenology; 2) Underestimation of groundwater recharge and release to streams during the growing season; and 3) A lack of household modulation of groundwater use during dry periods. During our prior surveys (Law et al., 2004; Fraser et al., 2013) residents stated they had reduced their water use during droughts. While the model underestimation was small (up to ~0.5 mm day$^{-1}$), additional empirical data about water flux, groundwater processes, and household water management would enhance model prediction accuracy of hydrological processes, especially during the growing season.

## 4.2  N dynamics and uncertainties

### 4.2.1    Nitrogen concentrations and loads

Not surprisingly, adding fertilization and septic modules in RHESSys improved the simulations of in-stream $NO_3^-$ concentration and load dynamics. Compared to the weekly BES observations, our model underestimated mean in-stream $NO_3^-$ concentration by less than 0.2 mg $NO_3^-$-N L$^{-1}$ (-10%) and with similar seasonality (Fig. 5). Considering that no N-related parameters were calibrated, the reasonable $NO_3^-$ simulations suggest the model can provide sufficient assessment of the effects of household water and nutrient management on N transport, transformation, and export in suburban watersheds when only discharge but no $NO_3^-$ observations are available. Highest bias of $NO_3^-$ concentration was found in summer during our study period, when our model might retain excessive N in the upland through denitrification and uptake and leave little transported to streams. In addition, we assumed identical N inputs for all households in BARN, but the actual fertilization and septic effluents may have considerable spatial, and temporal variations which could impact the N cycling and transport significantly. Specifically, we used the annual fertilization rate on lawns as 84 kg N ha$^{-1}$ from Law et al. (2004) in which the reported range of annual fertilization was from 10.5 to 369.7 kg N ha$^{-1}$. It is also important to note that BARN did have extensive agricultural activities for up to two centuries which may have resulted in accumulation of legacy N in the groundwater.

Compared to other RHESSys studies (e.g., Lin et al., 2015; Son et al., 2019; Tague et al., 2013), spinning up the model for 30 years may be insufficient to account for the export of this N from groundwater, which possibly caused the lower simulated mean $NO_3^-$ concentration compared to BES measurements. A longer spin up period (i.e., 500 years) was used in Lin et al. (2015) for a fully forested watershed. In our suburban watershed with larger inputs and shorter residence time of N, the spin up period could be shorter than a fully forested watershed, as evidenced by asymptotic C:N ratio after 30 years. Furthermore, we found the model yielded a stronger seasonality of N export, with simulated concentrations with fertilization and septic processes lower during the growing season but spiking right at the end of growing season. Uncertainty in RHESSys phenology may contribute to these differences, where errors in the prescribed end of the growing season caused quick mobilization of $NO_3^-$ into streams. The lower estimation of streamflow during the growing season could also increase residence time and retention, and reduce N export from uplands and groundwater to streams, causing the underestimation of $NO_3^-$ concentration and load in these periods. Lastly, we noted that the observations of weekly $NO_3^-$ from BES tended to avoid the highest flow

conditions, but our model simulated $NO_3^-$ under all weather conditions. Bias between our model simulation and the observations is unavoidably expected.

Another interesting finding is that the simulated mean $NO_3^-$ concentration from scenario *none* was significantly greater than the observed concentrations at POBR (Table 3) which is a reference of forest and pre-urbanization conditions of watersheds in the region. The higher estimated $NO_3^-$ concentrations in BARN could be explained by the land use difference between the two watersheds. Specifically, there are more impervious areas and lawns in the upland of BARN than in POBR which is dominated by strongly N-retentive oak-hickory forests (with the exception of a regional natural gas line cut with herbaceous vegetation), resulting in lower N uptake and higher N concentration (Table 3, scenario *none* vs. POBR). This result implies that, even in the absence of additional $NO_3^-$ input from human activities, the water quality in urban watersheds is unlikely to fully recover to pre-urbanization levels due to altered hydrology and differences in vegetation and land covers.

### 4.2.2    Denitrification and N retention hot spots

In addition to improving predictions of in-stream $NO_3^-$ concentration, the simulated denitrification rates (Sect. 3.3.2 & Table 4) in lawns fell in the range of empirically estimated rates at BARN (Suchy et al., 2023) and other areas in Baltimore (Raciti et al., 2011). Among all N retention hot spots, the constructed wetland and sediment accumulation zone at the base of the gully exhibited the highest denitrification rates within the entire watershed, both before and after considering fertilization and septic processes (Fig. A5). These rates were comparable to other wetland denitrification measurements: Groffman and Hanson (1997) estimated denitrification rates from 1 to >130 kg N ha$^{-1}$ year$^{-1}$ at several wetlands in Rhode Island; Poe et al. (2003) reported rates ranging between 19 to 191 kg N ha$^{-1}$ year$^{-1}$ at a constructed wetland receiving agricultural runoff; Harrison et al. (2011) found rates of 89 and 158 kg N ha$^{-1}$ year$^{-1}$ at two wetlands adjacent to Minebank Run in Baltimore. In BARN, these wetlands were located in low-slope downstream areas and advertently or inadvertently treat runoff originating from roads and upstream households. Unlike lawns which may not maintain high soil moisture levels, these areas remain consistently wet throughout most of the year. These features create ideal conditions for promoting denitrification and effectively retaining N loads that would otherwise be transported to streams. Specifically, these two wetlands covering only 0.09% of the watershed contributed to 0.39% of the total denitrification during the study period. This highlights the significance of strategically selecting locations for water quality improvement projects in future watershed restoration efforts, and assessing the ecosystem services of spontaneously generated features.

### 4.3  Future model improvements

The analyses here highlight several challenges in modelling ecohydrology of mixed land use watersheds such as BARN. Our current setup assumed a uniform daily $NO_3^-$ input and wastewater volume of septic effluents for all households and fixed fertilization amounts for lawns adjusted by application interval (Eq. 1). These parameters could be further adjusted when more observations are available. For fertilization, our model distributed the estimated total fertilization amount uniformly to all lawns in the watershed, at rates modulated by the proportion of lawns fertilized estimated by Law et al. (2004) and Fraser et

al. (2013). In reality, fertilization rate and frequency vary significantly in different lawns. Variable space and time patterns of fertilization rates could result in N input hot spots that exceed retention capacity relative to variable transport rates. For irrigation, our model applies irrigation close to its maximum (4 mm day$^{-1}$) when water stress is high, but residents may not irrigate their lawns at these rates during drought to conserve groundwater, and may continue to irrigate lawns during wet periods with automated sprinkler systems. Survey and high-resolution satellite observations could help to improve our irrigation module and accurately estimate the timing and quantity of irrigation practices in suburban watersheds. Current settings of our model could introduce excessive depletion of groundwater during droughts, and lead to underestimation of baseflow and in-stream $NO_3^-$ concentrations, or increased recharge during wet growing seasons. More detailed information about water use habits and observations of relationships between meteorological factors and groundwater storage are needed to improve simulation of the dynamics of water withdrawal in RHESSys.

## 4.4 Synthesis of results

Lastly, our study addressed three overarching questions:

1)    What are the individual and interacting contributions of different watershed N sources to streamwater N export?

Calibrating hydrologic parameters only, our augmented RHESSys model reduced the bias of $NO_3^-$ load (Table 3) significantly after including N loads of fertilization and septic effluents in BARN. Specifically, mean $NO_3^-$ load increased from 1.44 kg $NO_3^-$-N ha$^{-1}$ year$^{-1}$ in scenario *none* to 6.68 kg $NO_3^-$-N ha$^{-1}$ year$^{-1}$, compared to the 7.4 kg $NO_3^-$-N ha$^{-1}$ year$^{-1}$ observed export (Table 3). The reduced bias after adding human inputs showed our model could reasonably estimate the N export once the quantity and spatial patterns of N inputs are known. For BARN, the drop of retention rate in scenario *both* compared to scenario *fertilizer or septic only* suggested the watershed is saturated in its retention capacity, and there is a potential to promote N retention through new BMPs such as detention ponds and wetlands to reduce N export to streams through on- and off-site effects of hillslope hydrology and biogeochemistry.

2)    How do the spatially nested patterns of water and N inputs from human activities alter spatial patterns of a set of key ecohydrological processes including N retention, evapotranspiration, soil and groundwater levels and flows?

Simulation results indicate septic systems using deep groundwater as the water source, transport that water to shallow soils, resulting in systematic shallow water table increases within upland residential areas and small drops in water table levels in riparian areas of residential subcatchments. Results show how on-site extraction of water could alter the hydrological conditions of both "on-site" locations where septic effluent is directly disposed, as well as in "off-site" locations. These results occur because while the septic effluent is depleted by evapotranspiration, the deeper groundwater that emerges in riparian areas is also affected at hillslopes with residential development. Thus, extraction of water for domestic use lowers riparian water tables even when this water is ultimately discharged back into the environment via a septic system.  Likewise, the spatial pattern of denitrification showed increases not only in sites receiving N inputs directly (i.e., lawns and septic drainage fields) but also in "off-site" downstream areas (i.e., wetlands and riparian areas) receiving transported $NO_3^-$ from upland zones.

3) What are the patterns of hot spots for N retention and associated implications for the design of BMPs to promote N retention within suburban watersheds?

In the residential subcatchments of the watershed, riparian zones, constructed and accidental wetlands were found to be hot spots of denitrification (Zhang et al., 2023). These areas have the combination of subsidized supplies of water and $NO_3^-$, providing mixing zones with conditions promoting denitrification that are more consistent than fertilized lawn areas with variable soil moisture. Temporal patterns of denitrification were generally climate-driven with highest rates occurring in spring and summer in both hot spots and other areas in the watershed. Our results showed the spatial pattern of N retention and 580 identified spontaneously existing (accidental) retention zones that accumulate both water and N loads from upstream. By effective siting of BMPs based on our results for developed watersheds, both naturally occurring and built features could become N retention hot spots and provide ecosystem services to improve water quality in the future.

## 4.5 Conclusions

Our analysis provides important insights into how different sources of N input interact with ecohydrological processes to 585 control N export in suburban and exurban watersheds relying on local groundwater for domestic use and septic systems for wastewater release. With single-family houses dominant in these watersheds, the input of lawn fertilization and irrigation water, and septic effluent volume and N load are concentrated in limited areas at much higher per unit area rates. These differences cascade through the watershed producing hot spots of N export and retention. Calibrating hydrologic parameters against streamflow observations only, our model yielded satisfactory simulations of in-stream $NO_3^-$ concentration and upland 590 N retention processes. Specifically, our model estimated the mean $NO_3^-$ concentration as 1.43 mg $L^{-1}$, which is only less than 0.2 mg $L^{-1}$ lower than the weekly observations from Baltimore Ecosystem Study for our study period. The simulated denitrification rates at fertilized lawns are also comparable to measurements in the study area and nearby watersheds in Baltimore, and rates at wetlands and riparian areas are similar to reported measurements in other studies.

Our results strongly support the basis for small watershed-scale analysis and planning to address watershed N exports and are 595 particularly relevant in areas such as the Chesapeake Bay that are highly sensitive to N-induced eutrophication. Small-watershed improvement plans (e.g., Kamenetz, 2011) only have generic recommendations – more trees on lawns and reduce fertilizer inputs, without considering the spatial component of BMPs. The spatially explicit, high-resolution simulations from our model could help local decision makers to identify existing and potential new hot spots of N retention processes (e.g., denitrification) to further advance these plans. Specifically, we found locations accumulating both high N loads and water 600 from upstream are ideal locations for siting future BMPs (e.g., detention ponds, constructed wetlands) to promote N retention and improve water quality for local and downstream waterbodies. In summary, the improved RHESSys simulations with augmentations for more complete, spatially nested inputs of water and N and subsequent feedbacks between transport and retention highlight the importance of the structured spatial heterogeneity of human impacts to fully understand ecohydrological

processes at hillslope level in developed watersheds. Existing models often miss the patterns and feedbacks  water and N inputs at household levels and within hillslope hydrologic flowpaths. The spatially distributed inputs and our augmented RHESSys model structure may provide a reliable framework to comprehensively evaluate current coupled water, C and N cycles, and also understand and predict effectiveness of ecosystem restorations to improve water quality and ecosystem health in developed watersheds.



**Appendix A. Supplementary figures and tables**

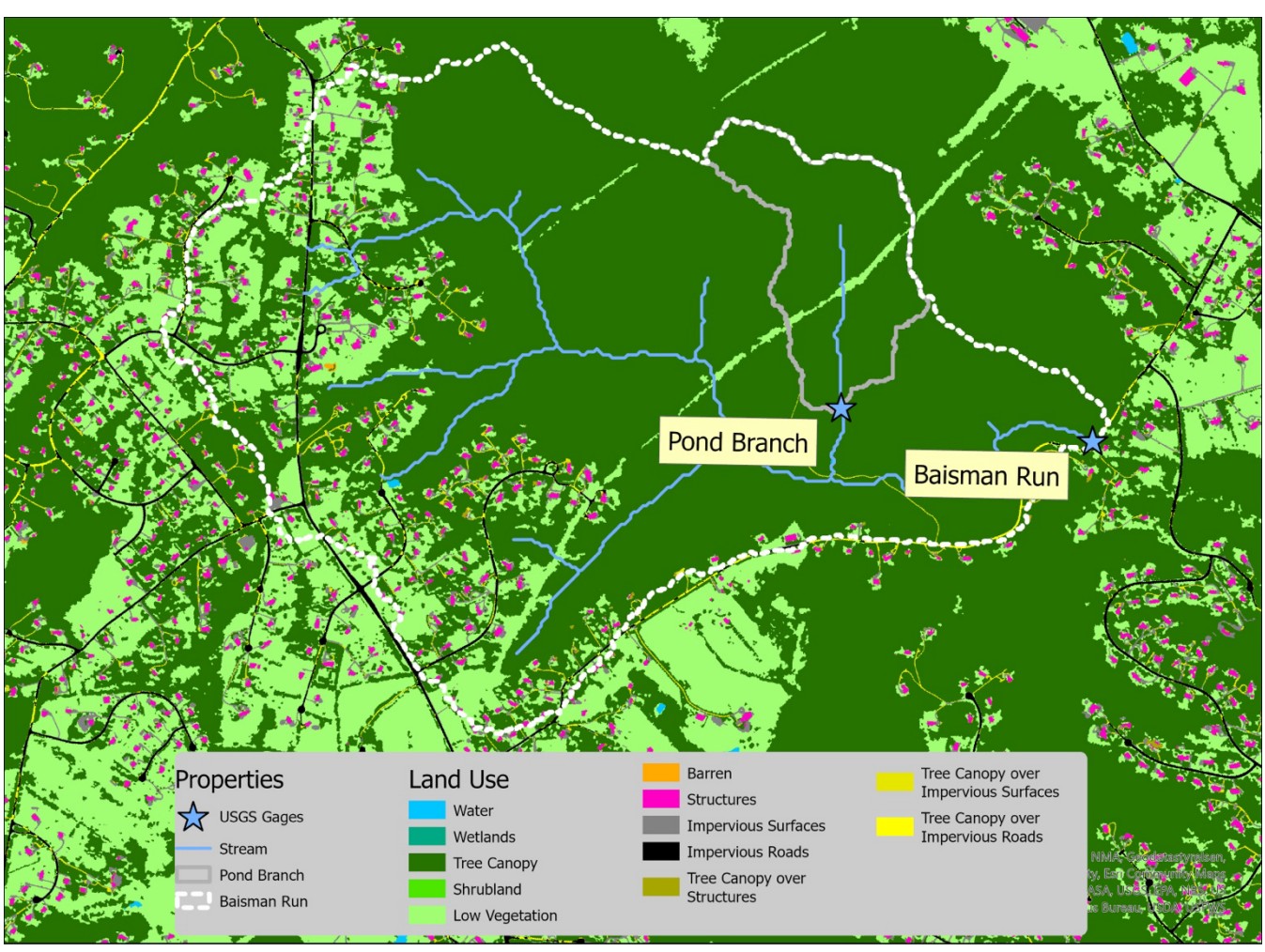

**Figure A1. 1-m land use and land cover in Baisman Run from the Chesapeake Bay Conservancy.**

**Table A1. List of data sources used to set up, calibrate, and evalute RHESSys model for Baisman Run**

| Data | Detail | Source |
|------|--------|--------|
| **Topography** | Bare Earth DEM 2014 | Baltimore County GIS, 2017 |
| **Land Use** | Chesapeake Bay 1-m Land Use | Claggett et al., 2018 |
| **Discharge** | United States Geological Survey | Gage ID: 01583580 (Baisman Run); 01583570 (Pond Branch) |
| **Water Chemistry** | Baltimore Ecosystem Study | Groffman et al., 2020; Castiblanco et al., 2023 |
| **Household Parcel** | Baltimore County Parcels | Baltimore County GIS, 2019 |
| **Hydrologic Network** | County Hydrolines | Baltimore County GIS, 2016 |

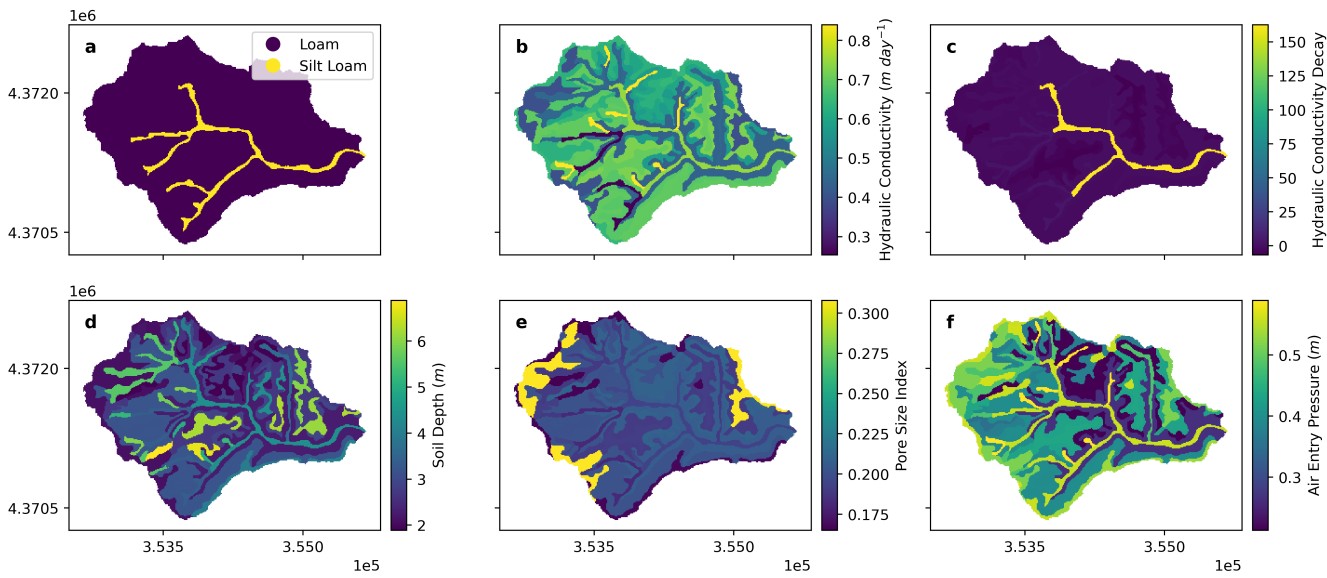


**Figure A2. SSURGO (USDA, 2019) derived (a) soil texture, (b) lateral and vertical saturated hydraulic conductivities at surface (m day⁻¹), (c) lateral and vertical decay rates for lateral and vertical hydraulic conductivities, (d) soil depth (m), (e) pore size index, and (f) air entry pressure (m) for Baisman Run.**

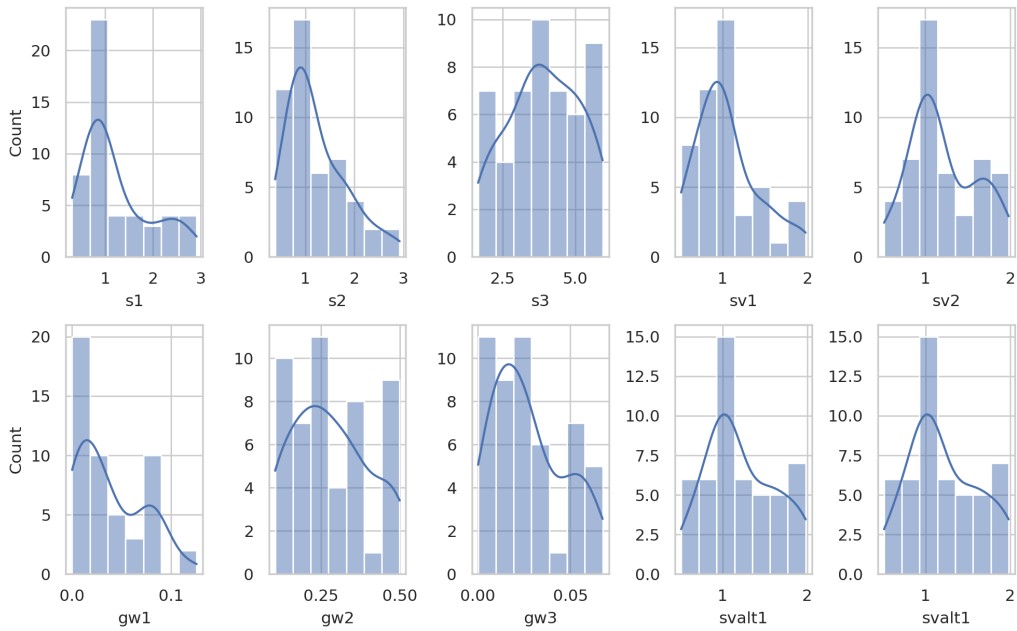


**Figure A3. Distributions of multipliers to RHESSys parameters (Table 1) based on 50 calibrated behavioral parameter sets.**

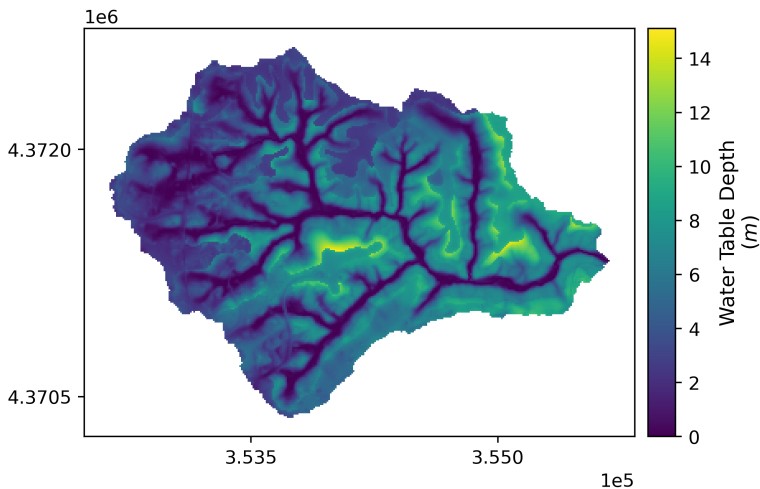

**Figure A4. Spatial pattern of ensemble mean water table depth (m) of Baisman Run during the entire study period (water year 2013 to 2017) from the 50 behavioral simulations under scenario *none*. Map in projection NAD83 UTM 18N (EPSG: 26918).**


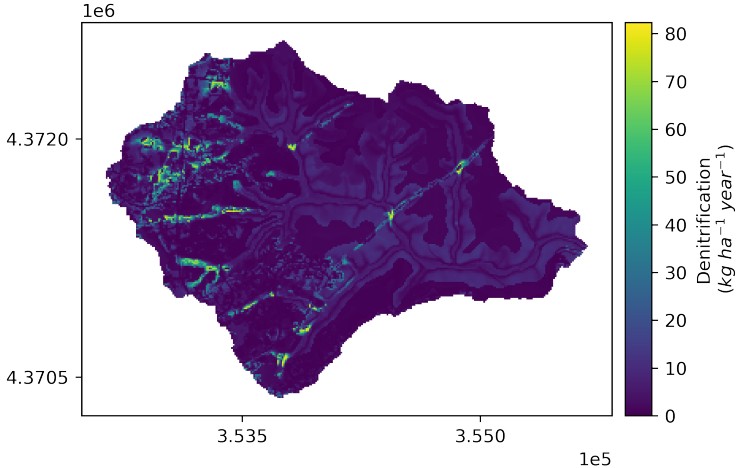

**Figure A5. Spatial pattern of ensemble mean denitrification (kg N ha⁻¹ year⁻¹) of Baisman Run during the entire study period (water year 2013 to 2017) from the 50 behavioral simulations under scenario *none.*. Map in projection NAD83 UTM 18N (EPSG: 26918).**

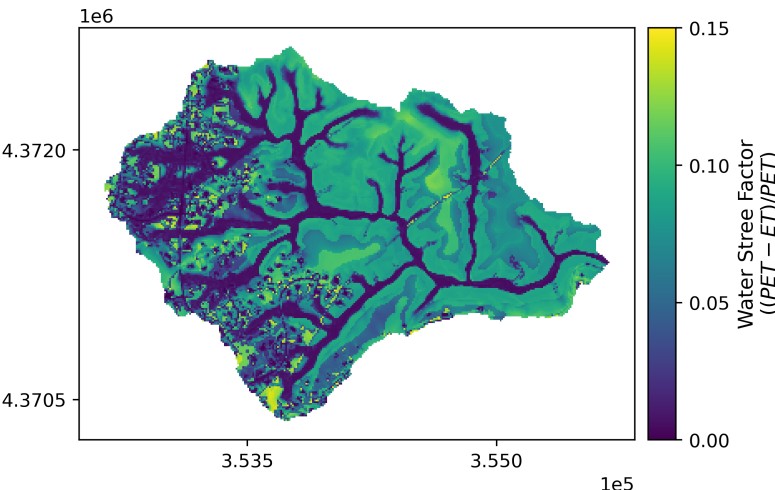

**Figure A6. Spatial pattern of ensemble mean water stress factor ((PET-ET)/PET, Eq. 3) of Baisman Run during the entire study period (water year 2013 to 2017) from the 50 behavioral simulations under scenario *none*. Map in projection NAD83 UTM 18N (EPSG: 26918).**

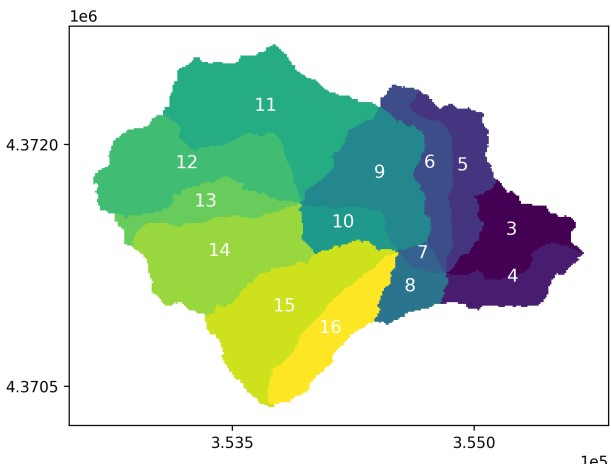

**Figure A7. Hillslope indices of Baisman Run. Map in projection NAD83 UTM 18N (EPSG: 26918).**

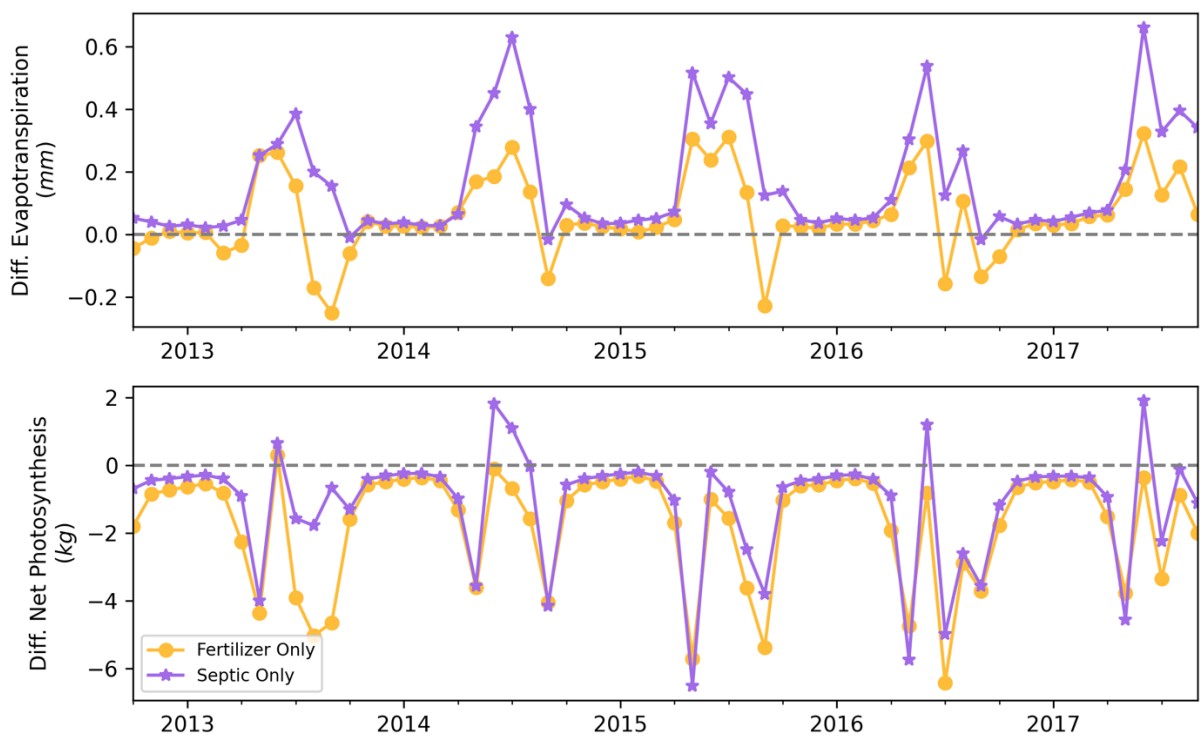


**Figure A8. Monthly differences of evapotranspiration (upper) and net photosynthesis (lower) between scenario *none* and scenarios *fertilizer only* and *septic only*.**

**Acknowledgements**

This work was supported by the National Science Foundation Coastal Science, Engineering, and Education for Sustainability Program, Grant No. 1426819, the National Science Foundation Long-Term Ecological Research (LTER) Program, Grant No. 1027188 for the Baltimore Ecosystem Study, and the Department of Energy Integrated Field Laboratory, Grant No. 004278 for the Baltimore Social-Environment Collaborative.

**Code and data availability**

The RHESSys program used for this study is available on https://github.com/ruoyu93/RHESSysEastCoast. The model definition files, outputs, and Python code used to simulate, analyse, and visualize the outputs (in Jupyter Notebook) are posted to a public Zenodo repository at https://doi.org/10.5281/zenodo.10022932 (Zhang et al., 2023). Other files related to the paper can be requested directly from the corresponding author (Ruoyu Zhang).

**Author contribution**

Conceptualization and main investigation, writing of the first draft, and visualization of this study was conducted by RZ under the supervision of LEB and PMG. LL provided technical assist on developing and calibrating the RHESSys model, and PMG, AKS, JMD, and AJG provided water chemistry and biogeochemical data. All authors reviewed and edited the paper.

**Competing interests**

The contact author has declared that none of the authors has any competing interests.

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
