# Peer review of "Simulation of spatially distributed sources, transport, and transformation of nitrogen from fertilization and septic system in a suburban watershed"

_Hydrology and Earth System Sciences, 2023_

## Referee Comment (RC2)

[referee-annotated manuscript omitted]

---

## Author Comment (AC1)

**Reviewer 1**

**Summary**
The work uses a calibrated fully-distributed ecohydrological model to explore the nitrogen sources, and the transport and transformation processes within a small exurban catchment. The manuscript seems to contribute to important process understanding, but the current presentation needs substantial revision to legible contribution to existing literature. My key concerns are: 1) lack of clarification on research novelty in the context of existing literature; 2) questionable capacity of the calibrated model to represent key processes, which raise further question on whether the model is appropriate to infer nitrogen dynamics from. I therefore suggest this manuscript to be returned to the authors for substantial revision and resubmission.

> Thanks for your helpful comments to our manuscript. We addressed your concerns to our study in novelty and model uncertainty for N dynamics accordingly in responds to your General Comments:
>
> We highlighted our revisions to the manuscript by using blue and calibri font with shading.
>
> - Updates are highlighted in blue
> - Responses in times new roman font

**General Comments**

1. What is the novelty of this study? A clear statement of this is needed with respect to the key knowledge gap in the existing literature: what is missing from current literature, and why are they important to consider. At present, the review of process-based modelling literature seems technically comprehensive, but it does not explain why the current study is needed as a useful addition to literature.

> Thanks for the comment. We highlighted the novelty of our study in the abstract and in the introduction and discussion. Our study addresses the **distribution and interaction of hillslope ecohydrological processes** in transporting natural and **human sources of nitrogen** in a long term monitored suburban watershed. Understanding processes and interactions at these scales promotes the design of retention features.
>
> To our knowledge, our model is the first fully distributed hydrologic model that includes i) spatial and temporal human-induced N and water sources at the household level, ii) hillslope ecohydrological processes for routing and cycling water, carbon, and nitrogen. These processes are necessary to identify the space/time distribution of "hot spots" of N retention at scales amenable to restoration.
>
> A significant aspect of the model is that it is calibrated for hydrologic processes restricted to soil and subsurface hydraulic parameters. It is not calibrated for biogeochemical processes which are subject to change with restoration activities. In contrast, the current

set of ecohydrological models typically calibrate patch (grid cells, elements) to stream transfer, and biogeochemical cycling parameters.

Abstract

Line 21:

We evaluated how the spatial and temporal distribution of nitrogen sources interacts with ecohydrological transport and transformation processes along surface/subsurface flowpaths to nitrogen cycling, and export. Embedding distributed household sources of nitrogen and water within hillslope hydrologic systems influences the development of both planned and unplanned "hot spots" of nitrogen flux and retention in suburban ecosystems.

Line 29:

With the model is calibrated for subsurface hydraulic parameters only and without calibrating ecosystem and biogeochemical processes, the model predicted mean […]

In the Introduction, we thoroughly reorganized the order of paragraphs and firstly highlighted why understanding ecohydrological processes at "hillslope level" is required for planning Best Management Practices and promote N retention.

Line 47:

BMPs can be both structural (e.g., constructed wetlands) and non-structural (e.g., changing fertilization and irrigation regimes). In addition to planned BMPs, spontaneously developed "hot spots" (Palta et al., 2017) may be responsible for a large share of nutrient retention, and therefore should be identified and protected. Both planned and unplanned retention features exist at very localized, sub-hillslope scales. Therefore, gaining a comprehensive understanding of the hillslope level ecohydrological behaviours and interactions between i) ecosystems and human derived nitrogen sources and ii) flowpath modification can lay the foundation for effectively mitigating these environmental issues through spatially well-conceived and sustainable management practices.

Then, we briefly reviewed how urban water quality is degraded by excessive human-induced N loads, emphasizing the widely used septic systems in suburban areas.\

Line 60:

In the United States, about 20% of households (26.1 million) are reported to be served by septic systems in 2007 (U.S. EPA, 2008). Through our work in Baltimore Ecosystem Study, low density suburban areas have been shown to produce the highest $NO_3^-$ load

per unit developed land among different land uses, degrading local and downstream water quality (Groffman et al., 2004; Zhang et al., 2022).

We then discussed the research gap in current semi-distributed models in aspects of incapable of including i) household scale human water and N loads contributing the majority of N inputs in suburban watersheds in distinct landscape positions and ii) hillslope hydrologic flow paths to meet the planning purposes to design BMPs to reduce N export. We also discussed data-driven approaches which could include additional N inputs, but hillslope-level N transport and transformation is still missing.

Line 69:

With rapid suburban and exurban sprawl, decision makers are facing environmental challenges which requires detailed planning for siting BMPs effectively in watersheds to promote N retention, reduce N export in streams, and protect water quality. These include both constructed and "inadvertent" biogeochemical hot spots at specific hillslope locations (e.g., swales, wetlands, riparian areas) on N retention at resolutions required for landscape design. However, commonly used modelling frameworks could not couple distributions and interactions of hillslope ecohydrological processes in transporting and transforming natural and human-induced N sources to understand or predict local (neighbourhood or hillslope) scale N transport and retention. Semi-distributed. Semi [...] lack(s) hillslope water and nutrient mixing along interacting surface/subsurface hydrologic flowpaths [...]

Line 82:
Data-driven approaches, such as SPARROW (Ator & Garcia, 2016; Smith et al., 1997), are also developed to assess large scale water quality in streams by nonlinear regression from gauged discharge and solute concentrations. However, these models also do not investigate hillslope-scale transport and transformation processes. In addition, there does not exist the data at hillslope scales to develop sufficient data-based approaches to understand and predict retention processes (e.g., denitrification, uptake, immobilization).

Then, we emphasized, though fully distributed hydrologic models, such as MIKE-SHE, could simulate hillslope hydrology and biogeochemistry, they currently have no modules to include the household-level N inputs developed.

Line 87:
Fully distributed hydrology models, such as MIKE-SHE (Abbott et al., 1986a, 1986b) and RTM-PiHM (Bao et al., 2017; Zhi et al., 2022), ParFlow (Maxwell, 2013) and RHESSys (Tague & Band, 2004) could explicitly couple hillslope hydrologic and biogeochemical processes that are required to understand transport and transformation of these human-induced N loads along hydrologic flowpaths from upland to stream.

Lastly, we wanted to highlight that our model is designed to be generalized to watersheds without long-term water chemistry observations which are quite expensive to acquire. In other words, we do not calibrate our parameters for N inputs (e.g., fertilization and septic loads) or processes but only soil hydraulics against streamflow records. If the model could reasonably estimate $NO_3^-$, it compromises the generalization of the model.

Line 102:

Lastly, the framework should be capable to be extrapolated to watersheds without water chemistry data which are less available than discharge records worldwide. It would be a valuable feature of the framework to estimate nutrient dynamics reasonably if calibrating only hydrologic parameters could provide reasonable estimation of N dynamics. Calibrating nutrient dynamics may not allow generalization to watersheds without chemistry records or extrapolation to conditions in which water quality BMPs are implemented.

2. The Introduction started discussion different types of models and their pros/cons from an earlier stage, lots of them are about inclusion of key processes (e.g., L55: hillslope water and nutrient mixing along hydrologic flowpaths). But for the readers' benefit, it might be clearer by adding a separate paragraph before introducing all the models, to discuss the theory about key processes at the particular spatial/temporal scale that you are interested in? Then you can start discussing and contrasting models based on their process representation.

Thanks for this suggestion. As in the response to Comment 1, we thoroughly reorganized the Introduction to improve its flow and readability. After the opening paragraph, we firstly emphasized the urgency to understand how excessive human N inputs affect water quality in urban watersheds, and then discussed the research gaps in current frameworks by comparing the semi- and fully distributed models and their limitations.

Lastly, we highlighted that our RHESSys model could be augmented to fill these research gaps in other models and advance our understanding to N dynamics of urban watersheds while recognizing some of the scale (watershed size) limitations.

Line 107:

The Regional Hydro-Ecological Simulator System (RHESSys, Tague & Band, 2004) is designed to meet all requirements for the framework, which is an ecohydrological model that simulates mass balances of water, C, and N of a watershed including hydrologic and biogeochemical stores and cycling. […]. In this study, we augmented RHESSys to include household-level transfer of groundwater for lawn irrigation and domestic water use, with domestic water use routed to septic spreading fields. With coupling hillslope hydrology and biogeochemistry at spatially connected patches, RHESSys could estimate spatiotemporal patterns of […] in spatially explicit manners. In summary, by adding modules of lawn irrigation, fertilization, and septic releases (see Sect. 2.3) that are commonly found in suburban areas, RHESSys is designed with the capacity to simulate the comprehensive ecosystem dynamics and feedbacks of introduced spatially explicit lawn irrigation, fertilization, and septic releases that are

commonly found in suburban areas, at resolutions commensurate with human management of the landscape. This facilitates scientific assessment of small-scale human activity and modification to land cover and infrastructure in expanding suburban and exurban areas.

3. You have a comprehensive review of process-based water quality models, what about the data-driven ones? The latter seem very useful to explain processes/changes at larger scales (e.g.,) – what's their relevance to your study? I think this comment can be potentially addressed once you have resolved my Comment #2.

Thanks for the comment. We addressed this in our response to Comment 1. Our model, compared to data-driven water quality models, is capable of providing the comprehensive representation of overall N cycling inside the watershed, which includes interacting processes (e.g., denitrification, uptake, nitrification, etc.) beyond $NO_3^-$ concentration at the outlet. Data-driven water quality models (e.g., SPARROW) may capture the change of stream N concentrations and loads due to land cover changes from urbanization within a watershed, but is not designed to estimate the impacts of small scale (below the level of a catchment) inputs of water and nitrogen and response to retention features. The data driven methods are useful for estimation of large-scale loads and concentrations of stream network N, but data to develop methods at the landscape scale we address are lacking. Therefore, we added a few sentences from line 108 to 112 contrast both approaches:

Line 82:
Data-driven approaches, such as SPARROW (Ator & Garcia, 2016; Smith et al., 1997), are also developed to assess large scale water quality in streams by nonlinear regression from gauged discharge and solute concentrations. However, these models also do not investigate hillslope-scale transport and transformation processes. In addition, there does not exist the data at hillslope scales to develop sufficient data-based approaches to understand and predict retention processes (e.g., denitrification, uptake, immobilization).

[Figure]

We discussed the details in Discussion that there are uncertainties in hydrologic behaviors and parameterization which could affect the simulation of $NO_3^-$ concentration, especially during the end of growing season (Fig. 3) when uncertainty of water usage and vegetation behaviors are not fully understood. Also, the spatial and temporal patterns of N inputs were assumed uniform for all households in the watershed, but the variations could significantly affect the N transport and transformation in the watershed. We also note that our observation samples were all collected under non-storm conditions, which could be quite different from our simulations which include all weather conditions. In summary, without calibrating N-related parameters of RHESSys, our model yield quite reasonable NO3- concentration compared to the observed records.

Line 483:

Considering that no N-related parameters were calibrated, the reasonable $NO_3^-$ simulations suggest the model can provide sufficient assessment of the effects of household water and nutrient management on N transport, transform, and export in suburban watersheds when only discharge but no $NO_3^-$ observations are available. The uncalibrated parameters of vegetation and domestic water usage introduced uncertainty in hydrologic and biogeochemical processes of our model, which may cause bias in streamflow and N cycling especially in the dry periods during the growing season. In these periods, our model might retain excessive N in the upland through denitrification and uptake, leaving little transported to streams. In addition, we assumed

identical N inputs for all households in BARN, but the actual fertilization and septic effluents may have considerable spatial, and temporal variations which could impact the N cycling and transport significantly. Specifically, we used the annual fertilization rate on lawns as 84 kg N ha$^{-1}$ from Law et al. (2004) in which the reported range of annual fertilization was from 10.5 to 369.7 kg N ha$^{-1}$. [...] Lastly, we noted that the observations of weekly $NO_3^-$ from BES were collected in conditions without large storm flows, but our model simulated $NO_3^-$ under various weather conditions. Bias between our model simulation and the observations is unavoidably expected.

6. There are some key information lacking in the Methods, some examples are listed below but they highlight need for a substantial improvement of the Methods section:

- Section 2.2 on calibration, was the model calibrated to only the streamflow record or with the water chemistry concentration data as well, and at which gauge? Please specify.

    Thanks for the comment. We highlighted in the responses to previous Comments that our calibration was performed only against the daily USGS discharge records, and no N-related parameters were calibrated. We added the USGS gage ID (Gage ID: 01583580) at line 216.

    Line 221:

    Lastly, we noted that no calibration was performed for N inputs (e.g., fertilization rate and septic load) or N cycling/transport processes in the model, as an important aim of our methods is to evaluate the capacity of our model to regionalize to watersheds where no water chemistry but only streamflow observations were available.

- In Table 1, what does the column 'sensitivity parameter' refers to? Also, for completeness, the table should also present the original parameter values estimated from SSURGO soils dataset besides the calibrated multipliers.

    Thanks for this comment. We included physical meanings of parameters in Table 1 and the original SSURGO values in Fig B2. The SSURGO values were estimated for each type of soils and varies among patches, therefore we could not include a single value for each parameter but showed the maps of these values in Fig. B2.

    We added a sentence for readers to check supplementary for more information about SSURGO soil at line 202.

    [...] we calibrated eight parameters (Table 1) for subsurface properties (i.e., lateral and vertical saturated hydraulic conductivities and their decay rates, pore size index, and air entry pressure) with initial estimates (Fig. A2) from the SSURGO soils dataset (USDA, 2019) and deeper groundwater processes (i.e., bypass seepage from surface and shallow saturated soil, and drainage rate to stream). [...]

[Figure]

Figure A2. Soil properties derived from SSURGO dataset. These values are calibrated against USGS observations and modified by multipliers listed in Table 1.

**Table 2. RHESSys parameters being calibrated and their physics (Tague and Band, 2004). Calibrated results shown as ranges of multipliers to original soil properties (Fig. A2 & A3) and groundwater component generating behavioural simulations with NSE greater than 0.5 for streamflow.**

| Parameter Groups | RHESSys Parameter Abbreviations | | Detail | Source | Unit | Multiplier Range |
|---|---|---|---|---|---|---|
| Lateral soil hydraulics | s | $m_l$ | Decay rate of lateral saturated hydraulic conductivity with depth | USDA SSURGO, 2019 | - | 0.31 – 2.91 |
| | | $K_{sat0\_l}$ | Lateral saturated hydraulic conductivity at the soil surface | | m day$^{-1}$ | 0.38 – 2.93 |
| | | $z$ | Soil depth | | m | 1.65 – 5.95 |
| Vertical soil hydraulics | sv | $m_v$ | Decay rate of vertical saturated hydraulic conductivity with depth | USDA SSURGO, 2019 | - | 0.51 – 1.98 |
| | | $K_{sat0\_v}$ | Vertical saturated hydraulic conductivity at the soil surface | | m day$^{-1}$ | 0.52 – 1.98 |
| Soil properties | svalt | $b$ | Pore size index | USDA SSURGO, 2019 | - | 0.51 – 1.98 |
| | | $\varphi_{ae}$ | Air entry pressure | | pounds inch$^{-2}$ | 0.5 – 1.05 |
| Groundwater dynamics | gw | $gw_1$ | Fraction of bypass from the saturated zone to groundwater storage | | - | 0 – 0.13 |
| | | $gw_2$ | Fraction of loss from groundwater storage to stream | | - | 0.03 – 0.5 |
| | | $gw_3$ | Fraction loss from surface to groundwater storage | | - | 0 – 0.07 |

- How are rainfall routing and runoff handled by the model? Are there any parameter to calibrated related to the rainfall-runoff processes?

  The rainfall-runoff processes of RHESSys are discussed in detail in Tague and Band (2004). At patch level, rainfall is intercepted by vegetation and infiltrated into its soil

layers. Surface and subsurface water is then routed to surrounding patches following hydraulic gradients. In subsurface, water is dynamically routed following gradients between water table elevations. Soil parameters, especially lateral and vertical soil hydraulic conductivity (i.e., **s** and **sv** in Table 1), affect the rainfall-runoff and drainage processes directly and are thus calibrated against the runoff observations. The multipliers will only alter the magnitudes of original SSURGO derived values (Fig. A2) but their spatial patterns are preserved. Soil hydraulic conductivities are assumed to decay exponentially, and the lateral and vertical decay rates (i.e., **m** in Table 1) are also calibrated to regulate water routing in this study. Surface routing features, including road and roof drainages, are also considered, as in Smith et al. (2022).

These parameters are commonly calibrated in previous RHESSys studies, and the routing procedure is detailed in Lin et al. (2021). The routing procedure of RHESSys is complex and well tested in previous studies (Smith et al., 2022). Therefore, to keep the focus of this study on N dynamics, we do not include the routing details in the Method, but provide the reference for readers to check at line 286:

RHESSys requires several subsurface hydraulic parameters to simulate lateral and vertical water flows and route subsurface lateral flow that are calibrated following the procedure detailed in Smith et al. (2022).

**References**

Smith, J. D., Lin, L., Quinn, J. D., & Band, L. E. (2022). Guidance on evaluating parametric model uncertainty at decision-relevant scales. *Hydrology and Earth System Sciences*, *26*(9), 2519-2539. https://doi.org/10.5194/hess-26-2519-2022

- Figure 2: why is rainfall not considered as a key process? How possible is lawn only irrigated by groundwater but not rain water?

  Rainfall is the most important water input of the watershed, and it included to all hydrological processes in RHESSys. The Fig. 2, however, is to highlight the new procedures of our augmentations for hillslope **groundwater redistribution** via. irrigation and septic systems, and these pumped waters were distributed to detention storage first and then follow the original RHESSys hydrological processes. Irrigation amount is regulated by the water stress in Equation 3.

  Line 251:
  Figure 2. Groundwater extraction for irrigation and septic systems in the RHESSys model. The source water (green arrow) is extracted from groundwater storage of drain-in patches (i.e., house centroids) and redistributed (orange arrow) to surface detention in downstream lawn patches for septic effluents and irrigated lawn patches of a household. After redistribution of source water, infiltration to soil and percolation to hillslope groundwater (yellow arrows) would follow the original processing of RHESSys

- Equation 3: PET and ET – how are there estimates?

  Potential evapotranspiration (PET) are estimated using the Penman-Monteith equation (Monteith, 1965) assuming no soil water limitations. PET representing the maximal ET

rate at given current meteorological information and land cover, and actual ET is estimated when the rate is regulated by soil moisture level and stomatal conductivity in each patch of our model. When water is not limited, PET and ET could be quite close; During droughts, PET could be much higher than the actual ET due to the low soil moisture level.

We provided the references to help reader refer for procedures and equations the RHESSys model uses to estimate PET and ET:

Line 286:

where *PET* and *ET* (mm) represent patch level potential and actual ET, which are estimated daily in RHESSys based on the Penman-Monteith equation (Monteith, 1965) and procedures in Section 5.6 in Tague and Band (2004).

7. The Results section presents a lot of information but there is no direct link of them to the modelling outputs. I think the Methods section misses a sub-section at the end on which model outputs are analysed and how, to answer which research question (which links to the Introduction). This would be very helpful for readers to link the Results section with the rest of the paper.

Thanks for the suggestion. We have first presented results in the Results section, but link those results to research questions in the Discussion section. We added a short paragraph in the end of our Methods Section 2.4 (line 327) to help readers refer to the corresponding sections in Results.

In the Results section, we presented model calibration results in Section 3.1, in-stream $NO_3^-$ dynamics of scenarios in Section 3.2, and ecohydrological changes and N hot spots in Section 3.3, accordingly.

**Specific Comments**

1. Line 21 – the statement seems too long and might be confusing, can you break this into two sentences, or use labels e.g., i), ii) if a single sentence is used?

Thanks for the comment. We broke down the sentence to align with other revisions, and changed the original sentence to:

Line 20:

We evaluated how the spatial and temporal distribution of nitrogen sources interacts with ecohydrological transport and transformation processes along surface/subsurface flowpaths to nitrogen cycling, and export. Embedding distributed household sources of nitrogen and water within hillslope hydrologic systems influences the development of

both planned and unplanned "hot spots" of nitrogen flux and retention in suburban ecosystems.

2. Figure 1 can be improved by including more information on the study area, including: locations of the two monitoring sites mentioned (01583580, 01583570) and the boundary of the sub-catchment, Pond Branch. The base map would be more informative presented as a map of key land uses (e.g., forest, urban, exurban) instead of a satellite image – it is a bit hard to visualize the land use components from the latter.

Thanks for the comment. I have added the USGS gages and the boundary of Pond Branch in the map as below.

[Figure]

We agreed the satellite image is a noisy background, and we replaced it with a general topographic map with hillshades outlined. The land use map contains 12 classes and could be too noisy for readers to view in the main manuscript, but we also added the land use map in the Appendix as Fig. A1 for readers who want to check the details of the watershed.

[Figure]

**Figure A1. 1-m land use and land cover in Baisman Run from the Chesapeake Bay Conservancy.**

3. Relating to my Comment #5, I'm also confused by your statement in L403 'our model underestimated the mean in-stream NO3 – concentration by 0.1 mg NO3 – -N/L (-7%) with stronger variability (Fig. 5)'. In Fig. 5, I see an approx. -50% bias comparing the simulated concentration for the 'both' scenario compared with the observation.

   Thanks for the comment. Please refer to our response to your Comment 4 and 5 for the details about the bias in the simulated $NO_3^-$.

---

## Author Comment (AC2)

**Reviewer 2**

**General Comments**

The manuscript entitled "Simulation of spatially distributed sources, transport, and transformation of nitrogen from fertilization and septic system in an exurban watershed" presents and uses an augmented version of the RHESSys Model to evaluate the hydrologic and biogeochemical N cycling, and transport in a mixed land use watershed characterized by anthropogenic N inputs from irrigation, fertilization, and on-site sanitary wastewater disposal in form of septic systems.

The study is motivated by enhancing the understanding of transport, cycling and subsequent export to streams of N in exurban watersheds. It declares the need to defer appropriate siting for effective best management practices (BMP) to reduce N export to downstream water bodies. To my perception, it fits within the thematic scope of HESS. It addresses a highly relevant research topic and could become a substantial contribution to scientific progress; however, the novelty of the presented approach remains unclear.

The manuscript promises to address certainly interesting aspects e.g., to evaluate how the spatial and temporal distribution of human nitrogen sources in exurban watershed controls N export to downstream water bodies and nitrification rates. However, the conclusions reached in the manuscript are rather general. At the current state, the concept of the study is limited to compare the simulated N loads and concentrations and nitrification rates between simulations without and with one or two human N input types and concludes that including fertilization and septic systems improves the simulation results when comparing to observations in the case study watershed. It remains unclear whether the augmented model can be transferred to other watersheds and how it can support to situate BMPs effectively.

The presentation of the results in the figures and tables does not keep up with the high quality in other articles published in HESS. Furthermore, to my understanding the methodology lacks significant steps: i) the model validation and ii) statistics that substantiate the results. Therefore, I suggest to reject the manuscript for publication at its current state and encourage the authors to improve it.

**Response:**

Thanks for your helpful comments to our manuscript. We addressed your concerns to our study in novelty and model uncertainty for N dynamics accordingly in responds to your General Comments:

We highlighted our revisions to the manuscript by using blue and calibri font with shading.

- Updates are highlighted in blue
- Responses in times new roman font

Thanks for all your insightful and helpful suggestions to improve the quality of our manuscript substantially. We addressed your three major concerns for 1) novelty of current approach, 2) too general conclusion, and 3) lack of model validation and statistics in methods accordingly.

1. **Unclear novelty**

    We highlighted the novelty of our study in the Abstract and Introduction. Our study addresses the **distribution and interaction of hillslope ecohydrological processes** in transporting natural and **human sources of water and nitrogen** in a long term monitored suburban watershed. Understanding processes and interactions at these scales promotes the design of retention features.

    To our knowledge, our model is the first fully distributed hydrologic model that includes i) spatial and temporal human-induced N and water sources at the household level, and ii) hillslope ecohydrological processes for routing and cycling water, carbon, and nitrogen. These processes are necessary to identify the space/time distribution of "hot spots" of N retention at scales amenable to restoration and Best Management Practices (BMPs) in the future.

    A significant aspect of the model is that it is calibrated for hydrologic processes **restricted to soil and subsurface hydraulic parameters**. It is **not calibrated for biogeochemical processes** which are subject to change with restoration activities. In contrast, the current set of ecohydrological models typically calibrate patch (grid cells, elements) to stream transfer, and biogeochemical cycling parameters. Therefore, our model could be generalized to other suburban watersheds with only discharge but no water chemistry observations.

    Abstract

    Line 21:

    We evaluated how the spatial and temporal distribution of nitrogen sources interacts with ecohydrological transport and transformation processes along surface/subsurface flowpaths to nitrogen cycling, and export. Embedding distributed household sources of nitrogen and water within hillslope hydrologic systems influences the development of both planned and unplanned "hot spots" of nitrogen flux and retention in suburban ecosystems.

    Line 29:

    With the model is calibrated for subsurface hydraulic parameters only and without calibrating ecosystem and biogeochemical processes, the model predicted mean […]

    In the Introduction, we thoroughly reorganized the order of paragraphs and firstly highlighted why understanding ecohydrological processes at "hillslope level" is required for planning Best Management Practices and promote N retention.

    Line 47:

BMPs can be both structural (e.g., constructed wetlands) and non-structural (e.g., changing fertilization and irrigation regimes). In addition to planned BMPs, spontaneously developed "hot spots" (Palta et al., 2017) may be responsible for a large share of nutrient retention, and therefore should be identified and protected. Both planned and unplanned retention features exist at very localized, sub-hillslope scales. Therefore, gaining a comprehensive understanding of the hillslope level ecohydrological behaviours and interactions between i) ecosystems and human derived nitrogen sources and ii) flowpath modification can lay the foundation for effectively mitigating these environmental issues through spatially well-conceived and sustainable management practices.

Then, we briefly reviewed how urban water quality is degraded by excessive human-induced N loads, emphasizing the widely used septic systems in suburban areas.

Line 60:

In the United States, about 20% of households (26.1 million) are reported to be served by septic systems in 2007 (U.S. EPA, 2008). Through our work in Baltimore Ecosystem Study, low density suburban areas have been shown to produce the highest $NO_3^-$ load per unit developed land among different land uses, degrading local and downstream water quality (Groffman et al., 2004; Zhang et al., 2022).

We then discussed the research gap in current semi-distributed models in aspects of incapable of including i) household scale human water and N loads contributing the majority of N inputs in suburban watersheds in distinct landscape positions and ii) hillslope hydrologic flow paths to meet the planning purposes to design BMPs to reduce N export. We also discussed data-driven approaches which could include additional N inputs, but hillslope-level N transport and transformation is still missing.

Line 69:

With rapid suburban and exurban sprawl, decision makers are facing environmental challenges which requires detailed planning for siting BMPs effectively in watersheds to promote N retention, reduce N export in streams, and protect water quality. These include both constructed and "inadvertent" biogeochemical hot spots at specific hillslope locations (e.g., swales, wetlands, riparian areas) on N retention at resolutions required for landscape design. However, commonly used modelling frameworks could not couple distributions and interactions of hillslope ecohydrological processes in transporting and transforming natural and human-induced N sources to understand or predict local (neighbourhood or hillslope) scale N transport and retention. Semi-distributed. Semi […] lack(s) hillslope water and nutrient mixing along interacting surface/subsurface hydrologic flowpaths […]

Line 82:
Data-driven approaches, such as SPARROW (Ator & Garcia, 2016; Smith et al., 1997), are also developed to assess large scale water quality in streams by nonlinear regression from gauged discharge and solute concentrations. However, these models also do not investigate

hillslope-scale transport and transformation processes. In addition, there does not exist the data at hillslope scales to develop sufficient data-based approaches to understand and predict retention processes (e.g., denitrification, uptake, immobilization).

Then, we emphasized, though fully distributed hydrologic models, such as MIKE-SHE, could simulate hillslope hydrology and biogeochemistry, they currently have no modules to include the household-level N inputs developed.

Line 87:
Fully distributed hydrology models, such as MIKE-SHE (Abbott et al., 1986a, 1986b) and RTM-PiHM (Bao et al., 2017; Zhi et al., 2022), ParFlow (Maxwell, 2013) and RHESSys (Tague & Band, 2004) could explicitly couple hillslope hydrologic and biogeochemical processes that are required to understand transport and transformation of these human-induced N loads along hydrologic flowpaths from upland to stream.

Lastly, we wanted to highlight that our model is designed to be generalized to watersheds without long-term water chemistry observations which are quite expensive to acquire. In other words, we do not calibrate our parameters for N inputs (e.g., fertilization and septic loads) or processes but only soil hydraulics against streamflow records. If the model could reasonably estimate $NO_3^-$, it compromises the generalization of the model.

Line 102:
Lastly, the framework should be capable to be extrapolated to watersheds without water chemistry data which are less available than discharge records worldwide. It would be a valuable feature of the framework to estimate nutrient dynamics reasonably if calibrating only hydrologic parameters could provide reasonable estimation of N dynamics. Calibrating nutrient dynamics may not allow generalization to watersheds without chemistry records or extrapolation to conditions in which water quality BMPs are implemented.

2. **Too general conclusion and model's ability to be transferred to other watersheds**
   We emphasized the major results of reasonable $NO_3^-$ concentration simulations, and the spatially explicit feature of our model allows assessments of BMPs' effects on promoting N retention when they are sited at areas **accumulating both high water and N loads** from upstream households in a watershed.

   Also, by performing uncertainty analysis, our $NO_3^-$ simulations include a reasonable range of biogeochemical outcomes while restricting calibration to subsurface hydraulic parameters. The model therefore can be applied to other sub/exurban watersheds which also use septic systems and fertilizers. In other words, with reasonable survey estimating human inputs and domestic water usage, our model could provide reasonable $NO_3^-$ export of watershed by calibrating against streamflow records which are much more available than the water chemistry data. For numerous suburban watersheds, our model could reasonably help decision makers understand the current N levels and upland dynamics without water chemistry data.

We thoroughly revised our Conclusion section:

Line 579:

Our analysis provides important insights into how different sources of N input interact with ecohydrological processes to control N export from suburban and exurban watersheds where single-family households use individual groundwater wells for domestic water discharged to septic systems and lawn irrigation, and add additional nitrogen in the form of sanitary effluent and lawn fertilization. While atmospheric deposition is ubiquitous, the input of lawn fertilization and irrigation water, and septic effluent volume and N load are concentrated in limited areas of the watershed at much higher per unit area rates. These differences cascade through the watershed producing hot spots of N export and retention. Calibrating hydrologic parameters against streamflow observations only, our model yielded satisfactory simulations of in-stream $NO_3^-$ concentration and upland N retention processes. Specifically, our model estimated the mean $NO_3^-$ concentration as 1.43 mg $L^{-1}$, which is only less than 0.2 mg $L^{-1}$ lower than the weekly observations from Baltimore Ecosystem Study for our study period. The simulated denitrification rates at fertilized lawns are also comparable to measurements in the study area and nearby watersheds in Baltimore, and rates at wetlands and riparian areas are similar to reported measurements in other studies. Our results strongly support the basis for small watershed-scale analysis and planning to address watershed N exports and are particularly relevant in areas such as the Chesapeake Bay that are highly sensitive to N-induced eutrophication. The spatially explicit, high-resolution simulations from our model could help local decision makers to identify existing and potential new hot spots of N retention processes (e.g., denitrification). Specifically, we found locations accumulating both high N loads and water from upstream are ideal locations for siting future BMPs (e.g., detention ponds, constructed wetlands) to promote N retention and improve water quality for local and downstream waterbodies. In summary, the improved RHESSys simulations with augmentations for more complete, spatially nested inputs of water and N and subsequent feedbacks between transport and retention highlight the importance of the structured spatial heterogeneity of human impacts to fully understand ecohydrological processes at hillslope level in developed watersheds. Existing models often miss the patterns and feedbacks water and N inputs at household levels and within hillslope hydrologic flowpaths. The spatially distributed inputs and our augmented RHESSys model structure may provide a reliable framework to comprehensively evaluate current coupled water, C and N cycles, and also understand and predict effectiveness of ecosystem restorations to improve water quality and ecosystem health in developed watersheds.

3. **Methods**

We appreciate your valuable suggestions to improve our Method section. We revised our approach to evaluate model outputs, expanded our discussion on the **model calibration and validation**, and **statistics** we used to quantify our model simulations in Results. Specifically, instead of showing the results from the simulation with highest NSE, we included more

simulations from parameter sets yielding NSE greater than 0.5 for our calibrating period with $gw_2$ parameter less than 0.5. We also note that no parameters for N inputs and related processes were calibrated in the study, aiming to evaluate whether the model could reasonably estimate $NO_3^-$ level by calibrating hydrologic parameters only. Please refer to **our response to Specific Comment #3 for details.**

From those behavioral simulations, we performed uncertainty analysis for streamflow and $NO_3^-$ concentrations, which strengthened the argument that our model is capable of simulating $NO_3^-$ dynamics without calibrating N related but only hydrologic parameters. With the updated method quantifying the uncertainty of our model, we updated our Results section substantially. Specifically, we composited simulations from 50 parameter sets with the **mean streamflow NSE from all simulations as 0.63 in the calibration period and 0.58 in the validation period**. For $NO_3^-$ concentration and loads, we showed that we resampled the daily simulation to weekly means, as our sample were collected only once a week under conditions without large storm flows. Without calibrating N processes, our ensemble mean from 50 parameter sets estimates the daily mean concentration of 1.43 mg $NO_3^-$-N $L^{-1}$, which is only 0.17 mg $NO_3^-$-N $L^{-1}$ lower than the observations in the study period.

We also thanks for your spotting of missing units in several equations, and we added units throughout all variables there.

Lastly, the detailed revisions are listed in our point-to-point responses to your suggestions in your attached PDF file (see Technical Corrections).

**Specific comments**

1. The motivation behind the study and the relevance of the research are well elaborated. However, the current state of the knowledge in regarding to the research questions is not elaborated. Are there no prior studies that have addressed similar research questions?

**Response:**

Thanks for the comment. To our knowledge, this is the first attempt to incorporate 1) spatial and temporal patterns of water and N inputs from irrigation, fertilizer, and septic systems in sub/exurban ecosystems and 2) evaluate their interactions with hillslope hydrologic and biogeochemical processes related to N retention. We reviewed several hydrologic and water quality models in the Introduction (in our response to 1. Unclear novelty), but found none include hillslope hydrology (i.e., explicit routing of water and nutrients within topography) and spatial and temporal patterns of N inputs from fertilizer and septic effluents simultaneously into one framework. Therefore, our augmented RHESSys model is by far the first fully distributed ecohydrological model that could meet the need to evaluate the current conditions of a watershed and designs of BMPs on forming hot spots for N retention.

2. The methodology should be written clearer and more structured. Given the fact that the study uses a rich base of data, for the reader it would be beneficial to have an overview of the data

used for setting up the model e.g., in the form of a Table providing specifications on each dataset and how it was employed in the study.

**Response:**
Thanks for the suggestion. We added a subsection to elaborate our calibration processes, and we also added a table in Appendix A to list all datasets we used for the study. We also changed our citation format thoroughly thanking to your suggestions.

**Table A1. List of data in Baisman Run used to set up model and analyze water chemistry**

| Data | Detail | Source |
|---|---|---|
| Topography | Bare Earth DEM 2014 | Baltimore County GIS, 2017 |
| Land Use | Chesapeake Bay 1-m Land Use | Claggett et al., 2018 |
| Discharge | United States Geological Survey | Gage ID: 01583580 (Baisman Run); 01583570 (Pond Branch) |
| Water Chemistry | Baltimore Ecosystem Study | Groffman et al., 2020; Castiblanco et al., 2023 |
| Household Parcel | Baltimore County Parcels | Baltimore County GIS, 2019 |
| Hydrologic Network | County Hydrolines | Baltimore County GIS, 2016 |

3. The model validation needs to be provided as well as the statistical methods for evaluating the simulation results.

**Response:**
Thanks for the suggestion. By summarizing your major comments to the Method section in the PDF file, we 1) reperformed model calibration and validation, 2) addressed why we chose the water year after 2010 to be evaluated in our study, 3) provided maps for initial values of soil properties from SSURGO in Supplementary (Fig. A2), which are further calibrated by multipliers to modify SSURGO properties' magnitudes but retain their spatial patterns.

**Model validation:**
We performed model calibration and validation again for our study, with the calibration period from water year 2013 to 2015 (Oct. 1, 2012 – Sep. 30, 2015) and validation period from water year 2016 to 2017 (Oct. 1, 2015 – Sep. 30, 2017). After calibration, we chose 50 behavioral parameter sets yielding highest NSE of streamflow to quantify uncertainty of model simulations using a 95% uncertainty boundary (i.e., 2.5$^{th}$ and 97.5$^{th}$ quantiles of simulations). The mean NSE of streamflow from the calibration period is 0.63 (range from 0.5 to 0.69), and 0.58 (range from 0.44 to 0.64) in the validation period. We noted that our calibration is only applied to hydrologic parameters of RHESSys, and no N-related parameters were calibrated.

Line 205:
We set the calibration period from water year 2013 to 2015 and validation period from water year 2016 to 2017. The original parameter values derived from SSURGO were further calibrated

by multipliers to vary their magnitudes but preserve the spatial patterns of soil hydraulic properties (Fig. A2). Specifically, the simulated streamflow was used to calibrate against the daily USGS discharge records (Gage ID: 01583580). From four thousands of parameter set realizations randomly chosen within specified limits, behavioural sets are chosen as yielding Nash-Sutcliffe efficiency (NSE; Nash & Sutcliffe, 1970) greater than 0.5 and fraction of groundwater loss to stream (i.e., gw2 in Table 1) less than 0.5 to estimate the ensemble means and uncertainties of model simulations. The latter condition was enforced to regulate the flashiness of groundwater dynamics, as BARN is found to have large saprolite storage to provide steady baseflow (Putnam, 2018). To assess uncertainty, we reported the 95% uncertainty boundaries for simulated streamflow and NO3- concentration and load from. Lastly, we noted that no calibration was performed for N inputs (e.g., fertilization rate and septic load) or N cycling/transport processes in the model, as an important aim of our methods is to evaluate the capacity of our model to regionalize to watersheds where no water chemistry but only streamflow observations were available.

**Why resample simulations from daily to weekly means?**
We compared the mean $NO_3^-$ concentration between the observations and model's weekly means. We resampled our concentration simulations because the direct comparison of daily model simulation and observation is difficult. RHESSys simulates the daily mean $NO_3^-$ under both low-flow and storm conditions, but our weekly grabbing samples were collected only in conditions with no large storm flows. Therefore, the weekly observations reflecting the average $NO_3^-$ level of the week, which is better compared to our model's weekly means, though the bias would be unavoidable in this way. Also, since no calibration was performed for N-related parameters, we reported results for the whole study period.

Line 314:

To better compare our $NO_3^-$ concentration results with the sampled weekly water chemistry from BES for BARN, we resampled the daily simulated concentration from RHESSys to weekly averages, expressed in unit of mg $NO_3^-$-N $L^{-1}$. The weekly $NO_3^-$ load was then estimated by the product of weekly mean $NO_3^-$ concentration and streamflow, expressed in unit of kg N $ha^{-1}$ $year^{-1}$. Note this approach may introduce bias for load as the once-a-week samples, typically not during major storms, and the observed daily mean discharges may not reflect the average load of the whole week.

Updated results for the validation period are shown in Table 3 with standard deviation reported from the means of $NO_3^-$ concentration and load from 50 simulations for each scenario.

Table 1. Mean weekly $NO_3^-$ concentration (mg N $L^{-1}$) and load (kg N $ha^{-1}$ $year^{-1}$) and corresponding standard deviation from calibrated simulations for BES weekly observations (BARN and POBR) and RHESSys simulation scenarios in each season and the entire study period from water year 2013 to 2017

| Variables | Season | Observation | | RHESSys Scenarios | | | |
|---|---|---|---|---|---|---|---|
| | | BARN | POBR | Both | Septic Only | Fertilizer Only | None |

| | | | | | | | |
|---|---|---|---|---|---|---|---|
| **Concentration (mg N L⁻¹)** | **Spring** | 1.5 | 0.02 | 1.4 (± 0.12) | 0.76 (± 0.08) | 0.77 (± 0.05) | 0.27 (± 0.03) |
| | **Summer** | 1.6 | 0.07 | 1.26 (± 0.13) | 0.68 (± 0.1) | 0.79 (± 0.1) | 0.33 (± 0.06) |
| | **Fall** | 1.57 | 0.06 | 1.41 (± 0.23) | 0.77 (± 0.15) | 0.94 (± 0.17) | 0.41 (± 0.09) |
| | **Winter** | 1.75 | 0.01 | 1.63 (± 0.18) | 0.88 (± 0.12) | 0.96 (± 0.1) | 0.35 (± 0.05) |
| | **Mean** | 1.6 | 0.04 | 1.43 (± 0.16) | 0.77 (± 0.11) | 0.87 (± 0.1) | 0.34 (± 0.06) |
| **Load (kg ha⁻¹ year⁻¹)** | **Spring** | 10.93 | 0.01 | 8.86 (± 0.63) | 4.84 (± 0.42) | 4.77 (± 0.31) | 1.62 (± 0.16) |
| | **Summer** | 5.88 | 0.02 | 4.72 (± 0.36) | 2.49 (± 0.25) | 2.81 (± 0.23) | 1.06 (± 0.16) |
| | **Fall** | 4.72 | 0.01 | 4.72 (± 0.39) | 2.57 (± 0.26) | 3 (± 0.27) | 1.23 (± 0.16) |
| | **Winter** | 8.38 | 0.01 | 8.42 (± 0.68) | 4.61 (± 0.46) | 4.91 (± 0.38) | 1.81 (± 0.18) |
| | **Mean** | 7.44 | 0.01 | 6.68 (± 0.47) | 3.63 (± 0.33) | 3.87 (± 0.27) | 1.44 (± 0.16) |

**Why water year after 2010?**

The carbon and nitrogen cycling in RHESSys generally required a long spin-up period to stabilize. In BARN, the developed areas in headwaters of used to be farmlands before 2000. After about 10 years of slow transformation on the farmland, no major development was found, and the land cover has been stable as the form when the land cover data was collected in 2013. To reduce the uncertainty of N inputs due to changes of number of households, land cover, and fertilization practices, we chose water year 2012 to 2017 to assess our model in a stationary condition. We discussed this at line 175:

BARN had gradual suburban development in the headwater which converted from agricultural land over a few decades. New development was largely completed in the 1990s, with one last field developed in 2007-2009. Our study period could reduce the uncertainty of N inputs due to land cover change during urban development and allow for analysis of N dynamics in a stationary condition.

4. The results need to include estimates of errors and indication of deviation between the analyzed years.

**Response:**

Thanks for your suggestions. We thoroughly revised our Results section to update our simulation results from the 50 behavioral simulations.

For streamflow, we updated our multipliers values in Table 1, and showed the standard deviation from all our simulations at Line 343:

In the calibration period (i.e., water year 2013 to 2015, Fig. 3a), the ensemble of simulated mean (standard deviation) daily streamflow was 1.24 ($\pm$0.03) mm day$^{-1}$, with NSE of 0.63 (between 0.5 and 0.69) compared to the USGS observed 1.38 mm day$^{-1}$. In the validation period (Fig. 3b), the simulated ensemble mean (standard deviation) streamflow was 0.91 ($\pm$0.03) mm day$^{-1}$, with NSE of 0.58 (between 0.44 to 0.64) compared to the USGS's 0.86 mm day$^{-1}$.

For $NO_3^-$ concentration/load, we reported the results of the ensembled mean value from 50 behavioral simulations in Table 3 (see above). Note we no longer reported the streamflow-weighted $NO_3^-$ concentration in the revised version, as the reported ensemble results could 1) better assess the uncertainty of our model simulations and 2) be directly compared with results in Figure 3. We updated our contents in Sect. 3.2, Line 368:

We calculated weekly means of $NO_3^-$ load and concentration of behavioural simulations. In our 5-year study period, the ensemble mean $NO_3^-$ concentrations (Fig. 4a) for scenarios *none, septic only, fertilization only, and both* were 0.34, 0.77, 0.87, and 1.43 mg $NO_3^-$-N L$^{-1}$, respectively (Table 4). The mean long-term observed concentration at the BARN USGS gauge was 1.6 mg $NO_3^-$-N L$^{-1}$. Thus, the simulated bias of mean $NO_3^-$ concentration considering both fertilization and septic loads decreased significantly from -1.26 mg $NO_3^-$-N L$^{-1}$ in the scenario *none* to 0.17 mg $NO_3^-$-N L$^{-1}$ in the scenario *both*. The 95% uncertainty boundary of weekly $NO_3^-$ concentration in scenario *both* captured 67% of the weekly sampled observations. The seasonality of $NO_3^-$ concentration is also well captured, except for the growing season (e.g., Jul. to Oct. in 2013 and 2016) when the model underestimated low flows (Sect. 3.1).

At line 394, we updated the ensemble results for water table depth, with a standard deviation of 1.1 m from 50 behavioral simulations. We also refined our results for the residential hillslopes (Fig. A6, hillslope 11 to 16) with urban development in BARN to see how human activities affect the ecohydrological behaviors.

The ensemble mean of water table depth (Fig. A4) from all behavioural simulations under scenario *none* was 4.52 m during the study period. Fertilization had overall negligible effects on watershed mean soil moisture or water table depth compared to the base (*none*) scenario (Fig. 6a – 6c), but minor increase of water table depth was detected in the residential areas, likely due to higher ET in lawns after fertilization. Septic processes decreased mean water table depth to 4.47 m by groundwater mounding, which increases shallow groundwater flow to surrounding patches along connected flowpaths. Specifically in septic drainage field patches, the mean water table depth decreased to 3.69 m (-0.66 m, -15%) in scenarios *both* and 3.72 m (-0.63 m, -14%) in *septic only* compared to the mean depth of 4.35 m, in scenarios *none* and *fertilization only*. With setting hillslope groundwater as the only source for septic process, we found groundwater withdrawal resulted in drier conditions (i.e., increase of water table depth) in riparian areas of these residential hillslopes (Fig. A6, hillslopes 11 to 16), where the mean water table depth increased by 5 (2%) and 8 (3.4%) mm in scenarios *septic only* and *both* compared to 219 mm depth in scenarios *none* and *fertilization only*. Though the **standard deviation** of each scenario from the 50 behavioural simulations was 1.1 m, the spatial distribution of soil moisture is consistent among all behavioural simulations.

We did the same for ET at line 406:

The watershed-scale mean ET was 43.9 mm month$^{-1}$ in scenario *none* and 44.0 mm month$^{-1}$ in scenario *fertilization only*. The standard deviation from 50 behavioural parameter sets was 0.8 mm month$^{-1}$ for each scenario. With septic processes activated, mean ET increased to 44.1 and 44.2 mm month$^{-1}$ in scenarios *septic only* and *both* in the residential hillslopes. [...]. With septic processes activated, mean ET increased to 44.1 and 44.2 mm month$^{-1}$ in scenarios *septic only* and *both* in the residential hillslopes, which could be contributed by the additional water extracted from groundwater to surface soil at the upland areas (in Fig. 6). When fertilization is activated in scenario *fertilization only*, ET in riparian areas of residential hillslopes decreased to (by) 54.7 (-0.1, -0.3%) mm month$^{-1}$ compared to scenario *none*, while the upland of these hillslopes increased by 0.1 mm month$^{-1}$. This showed that fertilization in the upland residential lawns could support higher growth rate of vegetation but preventing water from draining towards downstream areas of a hillslope (in Fig. 6).

As a respond to the soil moisture condition, the ensemble watershed mean denitrification rate dropped compared to our previous simulation using only one parameter set. We reported the new results thoroughly at line 419:

Compared to scenario *none* (Fig. A5), the ensemble mean annual rates of denitrification at the watershed scale were 7.2, 7.8, and 9.1 kg N ha$^{-1}$ year$^{-1}$ in scenarios *fertilization only*, *septic only*, and *both*, respectively, increasing by 33%, 44%, and 68% (Fig. 6d – 6f & Table 4). The standard deviation from the 50 behavioural simulations was 1.5 kg N ha$^{-1}$ year$^{-1}$ for scenario *none* and *fertilization only* and 1.6 kg N ha$^{-1}$ year$^{-1}$ for scenario *septic only* and *both*. When fertilization and septic processes were activated, the denitrification rates increased at the residential hillslopes and their riparian areas. The only exception was found in scenario *septic only*, where 7 patches experiencing minor reduced denitrification (-1.4% in average). All these patches were found in riparian areas of residential hillslopes where the water table drops by 9 mm in average after the septic processes extracting groundwater in the upstream.

Lastly, according to your suggestions to improve our maps, we integrated the previous figures for water table depth and denitrification by only showing the differences from our scenarios (Fig. 6), and move the original maps as supplementary (Fig. A)

[Figure]

**Figure 6. Ensemble mean differences of water table depth (top panel) and denitrification (lower panel) between scenario *none* and scenario fertilizer only (a & d), septic only (b & e), and both (c & f). The two hot spots of denitrification (i.e., wetlands in Fig. 1) were circled in (f).**

[Figure]

**Figure A4. Spatial pattern of ensemble mean water table depth (meter) of Baisman Run during the entire study period (water year 2013 to 2017) from the 50 behavioral simulations. Map in projection NAD83 UTM 18N (EPSG: 26918).**

[Figure]

**Figure A5. Spatial pattern of ensemble mean denitrification (kg N ha⁻¹ year⁻¹) of Baisman Run during the entire study period (water year 2013 to 2017) from the 50 behavioral simulations. Map in projection NAD83 UTM 18N (EPSG: 26918).**

[Figure]

**Figure A6. Hillslope indices of Baisman Run. Map in projection NAD83 UTM 18N (EPSG: 26918).**

5. Some figures (maps) are obsolete as to my opinion the difference between them can not be spotted. Some figures have confusing axis labeling, lack titles for the legends and some have unclear captions (see technical corrections file for more details).

**Response:**

Thanks for your suggestions to improve our figures. According to your suggestions, the labels, legends, and captions of all figures are rephrased/fixed. Please refer to the end of this documents to see the updated figures. We made a new figure (Fig. 6, see above) to highlight the differences of water table depth and denitrification between scenario *none* and other three human N scenarios, since we agreed that the maps in the left panels showing absolute values are difficult to be differentiated. The spatial patterns of water table depth of scenario none were provided in Fig. A4 and A5 (see above). Other updated figures are shown here:

[Figure]

**Figure 1. Study watershed Baisman Run (BARN) in suburban Baltimore County, Maryland (from ESRI). The black box highlights two N retention "hot spots": A sediment accumulation zone (upper circle) receiving drainage from roads and a constructed wetland (lower circle). These areas have a high capacity to prevent N from upland residential areas from being transported to streams.**

[Figure]

**Figure 2. Groundwater extraction for irrigation and septic systems in the RHESSys model. The source water (green arrow) is extracted from groundwater storage of drain-in patches (i.e., house centroids) and redistributed (orange arrow) to surface detention in downstream lawn patches for septic effluents and irrigated lawn patches of a household. After redistribution of source water, infiltration to soil and percolation to hillslope groundwater (yellow arrows) would follow the original processing of RHESSys**

[Figure]

**Figure 3. The ensemble mean of daily streamflow from simulations (red) with NSE greater than 0.5 and USGS observations (blue), with the daily 95% uncertainty range from 50 simulations in grey for the (a) calibration (Oct. 2012 – Sep. 2015) and (b) validation (Oct. 2015 – Sep. 2017) period. All simulations turned on irrigation, lawn fertilization, and septic processes**

[Figure]

**Figure 4. Ensemble weekly mean (a) NO₃⁻ concentration and (b) load at the outlet of Baisman Run over the entire study period (water year 2013 to 2017). The 95% uncertainty boundary for scenario *both* was shown in grey.**

6. The manuscript needs to be reviewed i) to comply with the HESS requirements for manuscript composition, ii) for unit formatting as required by the submission guidelines, iii) for correct referencing of used data sets and software

**Response:**

Thanks for the comment. We have fixed our unit format. The hyperlinks for datasets have been fixed using the correct reference style according to HESS guidelines.

**Technical corrections**

Please find all technical corrections and suggestions for improvement of the figures as comments in the attached manuscript.

**Response:**

Thanks for all the corrections and suggestions your made to your manuscript. We listed major/important comments you have here for your reference. Corrections to typos are made in the manuscripts accordingly.

o   Line 16: consistent terminology in abstract. Here you chose exurban and later in the abstract suburban.

Thanks for the suggestion. We think suburban is the more general terminology for the study. Though BARN is not a typical suburban watershed, we noted it could be treated as a low-density suburban watershed, which is exchangeable with "exurban" for Baisman Run in the abstract.

Excess export of reactive nitrogen in the form of nitrate ($NO_3^-$) from suburban watersheds is a major source of water quality […]. These processes in turn control the development of "hot spots" of nitrogen flux and retention in suburban ecosystems. We chose a well-monitored low-density suburban or exurban watershed, Baisman Run in Baltimore County […]

o   Line 47: is this about the spatial distribution of them?

Thanks for the question. Yes, this is about the spatial planning of the management practices. We rewrote here as:

Line 54:
Therefore, […] for effectively mitigating these environmental issues through spatially well-conceived and sustainable management practices.

o   Line 93: because of the biogeochemical modules?
Thanks for the question. RHESSys could simulate these detailed ecohydrological processes because it simulates fully distributed hillslope hydrology and coupled C and N dynamics in soil interacting with water and vegetations. We revised this sentence as at line 113:

In this study, we augmented RHESSys to include household-level transfer of groundwater for lawn irrigation and domestic water use, with domestic water use routed to septic spreading fields. With coupling hillslope hydrology and biogeochemistry at spatially connected patches, RHESSys could estimate spatiotemporal patterns of soil moisture, lateral flow distribution, evapotranspiration, groundwater level, and N transportation, transformation, uptake, and immobilization in spatially explicit manners.

o   Line 107: (The third research question) It's not very clear.

Thanks for the comment. We agree the third research question could be further clarified as below at Line 130:

What are the patterns of hot spots for N retention and associated implications to design future BMPs to promote N retention within suburban watersheds?

o   Line 140: This sentence contains redundant information from L130, you can combine it.

Thanks for the suggestion. We have removed this sentence.

o Line 149: at which basis?

Thanks for the comment. The atmospheric deposition of N was observation records from the National Atmospheric Deposition Program (NDAP) site MD99 (https://nadp.slh.wisc.edu/sites/ntn-MD99/). We added the reference at Line 170:

Atmospheric N deposition was estimated as 11 kg N/ha/year from site MD99 of National Trends Network from National Atmospheric Deposition Program (NADP, 2022).

o Line 151: Could you elaborate why you chose this 5 year period out of the available data (that I understood to cover 2000-2018)?

Thanks for the question. This is because there was continuous urban development before 2012 in BARN. The land use data were also acquired in 2013 and would not berepresentative to the conditions before it. We therefore chose study period after 2012 to make sure the stationarity of land over and excluded N loads uncertainties. We answered this in detail in the response to Specific Comment #3.

Line 175:

BARN had gradual suburban development in the headwater which converted from agricultural land over a few decades. New development was largely completed in the 1990s, with one last field developed in 2007-2009. Our study period could reduce the uncertainty of N inputs due to land cover change during urban development and allow for analysis of N dynamics in a stationary condition.

o Line 163: could you provide the number of houses this to quantify the stated uncertainty

Thanks for the suggestion. We added this number and rewrote the sentence at Line 187:

We identified 181 households, although 13 homes are located on the watershed divide, providing some uncertainty to the effective number of septic systems.

o Line 170: Did you perform this sensitivity analyses?

Thanks for the question. We did test to set the starting date early and late (i.e., 35 days ahead and backward) for grass (the LAI would stay high for longer period), but found negligible changes in water and N dynamics for BARN. As this is beyond the scope of this study, we removed this sentence to avoid confusion for readers.

o Line 174: The initial estimated should be listed together with the calibrated multipliers in Table 1

Thanks for the suggestion. The maps of initial values of SSURGO soil properties are added as Fig. A2.

[Figure]

**Figure A2. Soil types (a) based on SSURGO classification (USDA, 2019) and associated (b) lateral and vertical saturated hydraulic conductivities at surface (m day⁻¹), (c) lateral and vertical decay rates for lateral and vertical hydraulic conductivities, (d) soil depth (m), (e) pore size index, and (f) air entry pressure (pounds inch⁻²).**

We also updated our Table 1 to show ranges of multipliers applied on original SSURGO values.

**Table 2. RHESSys parameters being calibrated and their physics (Tague and Band, 2004). Calibrated results shown as ranges of multipliers to original soil properties (Fig. A2 & A3) and groundwater component generating behavioural simulations with NSE greater than 0.5 for streamflow.**

| Parameter Groups | RHESSys Parameter Abbreviations | | Detail | Source | Unit | Multiplier Range |
|---|---|---|---|---|---|---|
| Lateral soil hydraulics | s | $m_l$ | Decay rate of lateral saturated hydraulic conductivity with depth | USDA SSURGO, 2019 | - | $0.31 - 2.91$ |
| | | $K_{sat0\_l}$ | Lateral saturated hydraulic conductivity at the soil surface | | m day⁻¹ | $0.38 - 2.93$ |
| | | $z$ | Soil depth | | m | $1.65 - 5.95$ |
| Vertical soil hydraulics | sv | $m_v$ | Decay rate of vertical saturated hydraulic conductivity with depth | USDA SSURGO, 2019 | - | $0.51 - 1.98$ |
| | | $K_{sat0\_v}$ | Vertical saturated hydraulic conductivity at the soil surface | | m day⁻¹ | $0.52 - 1.98$ |
| Soil properties | svalt | $b$ | Pore size index | USDA SSURGO, 2019 | - | $0.51 - 1.98$ |
| | | $\varphi_{ae}$ | Air entry pressure | | pounds inch⁻² | $0.5 - 1.05$ |
| Groundwater dynamics | gw | $gw_1$ | Fraction of bypass from the saturated zone to groundwater storage | | - | $0 - 0.13$ |
| | | $gw_2$ | Fraction of loss from groundwater storage to stream | | - | $0.03 - 0.5$ |

| | | | | |
|---|---|---|---|---|
| $gw_3$ | Fraction loss from surface to groundwater storage | | - | $0 - 0.07$ |

o Line 179: How was the calibrated model validated?

    Thanks for the question. Please refer to our response to Specific Comment #3: Model calibration and validation.

o Line 180: The initial parameters and their units should be included in the table and the readers would profit from stating the Nash-Sutcliffe value here in the caption. Further, provide the meaning/names of the sensitivity parameters: s, sv, svalt, gw

    Thanks for your suggestions. We added a supplementary Fig. A2 (as above) to show the initial SSURGO values for each location of BARN, and improved our Table 1 as above.

o Line 212: Please add a (septic) reference if published.

    Thanks for the comment. We used data from Gold et al. (1990) and Lowe et al. (2009) to estimate the septic water and N load. The revised sentence at Line  is:

    We estimated the N load from septic systems as 7.7 kg N capita$^{-1}$ year$^{-1}$ and water input as 110.5 m$^3$ capita$^{-1}$ year$^{-1}$ (~80 gal$^{-1}$ capita$^{-1}$ day$^{-1}$), resulting in a $NO_3^-$ concentration of 70 mg N L$^{-1}$ estimated from results of Gold et al. (1990), Lowe et al. (2009), and other sources for per capita water use and septic nitrogen concentrations.

o Line 248: Is this the local practice in the study area?

    Thanks for the question. This 4 mm day$^{-1}$ threshold was set arbitrarily to constrain the groundwater extraction for septic or irrigation no more than this limit. Though we do not know the exact water extractions from each household, this limit allows abundant water usage that meets domestic water demand every day, and we did not see irrigation is beyond this limit during our study period assuming each house has 3.3 persons in average.

o Line 255: The survey by Law et al. (2004) and Fraser et al. (2013)?

    Thanks for the comment. We **removed** the sentence here to say it is consistent with survey results. We tried to include the maximal distance that people might irrigate their lawns, but there could be a quite large variations of this practice household by household.

o Line 257: Which method to you use to determine the differences between the scenarios?

    Thanks for the question. We compared the difference between the ensemble mean of $NO_3^-$ concentrations from 50 simulations for each scenario (Fig. 4). As discussed above, the direct comparison from our simulated daily average NO3- concentration and weekly

samples is difficult, we do not use traditional approaches (e.g., RMSE or $R^2$) in this study.

o Line 262: Could you elaborate on the method (resample daily to weekly)?

Thanks for the question. We answered this in our response to Specific Comment #3.

Section 2.4
To better compare our $NO_3^-$ concentration results with the sampled weekly water chemistry from BES for BARN, we resampled the daily simulated concentration from RHESSys to weekly averages, expressed in unit of mg N $L^{-1}$. The weekly $NO_3^-$ load was then estimated by the product of weekly mean $NO_3^-$ concentration and streamflow, expressed in unit of kg N $ha^{-1}$ $year^{-1}$. Note this approach may introduce bias for load as the once-a-week samples and the observed discharges at collecting days may not reflect the average load of the whole week.

o Line 273: I assume this subsection should be entitled Model calibration

Thanks for the suggestion. We changed this heading to "Model calibration and validation on streamflow". We also modified the axis titles in Figure 3.

[Figure]

**Figure 4. The ensemble mean of daily streamflow from simulations (red) with NSE greater than 0.5 and USGS observations (blue), with the daily 95% uncertainty range from 50 simulations in grey for the (a) calibration (Oct. 2012 – Sep. 2015) and (b) validation (Oct. 2015 – Sep. 2017) period. All simulations turned on irrigation, lawn fertilization, and septic processes.**

o Line 282: Was this (underestimation of streamflow in growing season) the same in every modelled year? Please provide at least a standard deviation and consider elaborating on the results for every year.

Thanks for the suggestion. With the uncertainty analysis, our 95% uncertainty range in Fig. 3 showed, most behavioral simulations would underestimate low flows in growing season (e.g., in 2013, 2016, and 2017). This is also found in previous studies in Baltimore (Miles, 2014). We discussed this at line 471:

This may be due to local increases in septic water and nutrients increasing ET during the growing season, reducing groundwater recharge, lowering groundwater storage, and reducing watershed baseflow. We also noted that our model tended to underestimate the lowest streamflows during the growing season, which was also found in another suburban watershed, Dead Run, in Baltimore by Miles (2014).

**Reference**

Miles, B. C. (2014). *Small-scale residential stormwater management in urbanized watersheds: A geoinformatics-driven ecohydrology modeling approach* (Doctoral dissertation, The University of North Carolina at Chapel Hill).

o Line 313: Please provide the standard deviations for NO3- concentration and load

Thanks for the comment. We included the standard deviations of $NO_3^-$ concentration and load in Table 3 (included in Specific Comment #3).

o Line 341: Please provide deviations with mean values

Thanks for the suggestion. We added the standard deviation of annual mean denitrification among water years of our study period for each scenario.

Compared to scenario *none* (Fig. A5), the ensemble mean annual rates of denitrification at the watershed scale were 7.2, 7.8, and 9.1 kg N ha$^{-1}$ year$^{-1}$ in scenarios *fertilization only*, *septic only*, and *both*, respectively, increasing by 33%, 44%, and 68% (Fig. 6d – 6f & Table 4). The standard deviation from the 50 behavioral simulations was 1.5 kg N ha$^{-1}$ year$^{-1}$ for scenario *none* and *fertilization only* and 1.6 kg N ha$^{-1}$ year$^{-1}$ for scenario *septic only* and *both*.

o Line 381: Discussions and conclusion should be separate sections.

Thanks for the suggestion. We have split out Discussion and Conclusion into two sections.

o   Line 402: Please substantiate your discussion points with references. Do other studies using RHESSys encounter similar issues?

Thanks for the question. The underestimation of low flows during growing season was also detected in previous RHESSys studies for another suburban watershed, Dead Run, in Baltimore by Miles (2014) at line 471.

We also noted that our model tended to underestimate the lowest streamflows during the growing season, which was also found in another suburban watershed, Dead Run, in Baltimore by Miles (2014).

o   Line 405: Please quantify the uncertainties (septic and fertilization)

Thanks for the suggestions. We rephrased the sentence, emphasizing that each household has different fertilization or septic release rates and the spatial variation of N inputs could affect the N transport and transform and our model simulations. However, the actual rates of N inputs from fertilization and septic systems for all households is quite challenging to estimate at this point. We therefore, reported the range of surveyed fertilization rate from Law et al. (2004) to show the input variations.

Line 490:

In addition, we assumed identical N inputs acquired from Law et al. (2004) for all households in BARN, but the actual fertilization and septic effluents may have considerable spatial, and temporal variations which could impact the N cycling and transport significantly. Specifically, we used the annual fertilization rate on lawns as 84 kg N ha$^{-1}$ from Law et al. (2004) in which the reported range of annual fertilization was from 10.5 to 369.7 kg N ha$^{-1}$.

o   Line 408: Please substantiate with references. How many spin up years were used in other studies?

Thanks for your suggestion. We added other studies for RHSSys, which used 500-year (Lin et al., 2015), 82-year (Son et al., 2019), or 47-years (Tague et al., 2013) spin-up periods to stabilize the model.

Line 495:
Compared to other RHESSys studies (e.g., Lin et al., 2015; Son et al., 2019; Tague et al., 2013), spinning up the model for 30 years may be insufficient to account for the export of this N from groundwater, which possibly caused the lower simulated mean NO$_3^-$ concentration compared to BES measurements.

**References**
Lin, L., Webster, J. R., Hwang, T., & Band, L. E. (2015). Effects of lateral nitrate flux and instream processes on dissolved inorganic nitrogen export in a forested catchment: A model sensitivity analysis. *Water Resources Research*, *51*(4), 2680-2695.

Son, K., Lin, L., Band, L., & Owens, E. M. (2019). Modelling the interaction of climate, forest ecosystem, and hydrology to estimate catchment dissolved organic carbon export. *Hydrological Processes*, *33*(10), 1448-1464.

Tague, C. L., Choate, J. S., & Grant, G. (2013). Parameterizing sub-surface drainage with geology to improve modeling streamflow responses to climate in data limited environments. *Hydrology and Earth System Sciences*, *17*(1), 341-354.

- o Line 423: Please compare the rates to the specific rates from the references like done for the hot spots in the following paragraph

  Thanks for the suggestion. The denitrification rate at lawn was measured in lab with fixed environment settings (in Line 446, the Result section).

  Assuming 210 days (~7 months) that denitrification would occur, Raciti et al. (2011) reported a denitrification rate of 204 kg N ha$^{-1}$ year$^{-1}$ at 20 °C for saturated soil samples from fertilized lawns at the University of Maryland Baltimore County. At the same temperature, Suchy et al. (2023) reported a higher rate, 744 kg N ha$^{-1}$ year$^{-1}$, when lawn soil samples collected from BARN lawns were saturated.

  However, direct conversion of lab measured rates to the field measurements is impossible as the environment variables change all the time. We used Raciti et al.'s (2011) approach, the estimated denitrification rates were 13 and 40 kg/ha/year, respectively, using measurements from Raciti et al. and Suchy et al. These values were reported at Line 450. We added the cross reference to let readers to check the estimated rates.

  Line 452:

  The mean 25 and 85 percentiles of annual denitrification rate for lawns from all simulations in scenario *both* were 2.8 to 30.8 kg N ha$^{-1}$ year$^{-1}$, respectively, which are quite comparable with the range of empirical measurements from low to high soil moisture conditions and various fertilization rates.

- o Line 467: What does unaffected mean?

  Thanks for your question. Our updated results suggested there was negligible change of water table depth at riparian areas at the whole watershed scale, but the drop of groundwater due to septic extraction is significant at hillslopes with dense residential development. We revised this sentence to explain this at line 562:

  These results occur because while the septic effluent is depleted by evapotranspiration, the deeper groundwater that emerges in riparian areas is also affected at hillslopes with residential development. Thus, extraction of water for domestic use lowers riparian water tables even when this water is ultimately discharged back into the environment via a septic system.

o Line 478: Please specify where BMPs are sited effectively in a watershed. It would be interesting to run simulations with additional BMPs or BMPs in different locations throughout the watershed and compare those.

Thanks for the comments. We mentioned that areas accumulating both upstream water and N inputs are ideal sites for BMPs. Running scenarios of siting BMPs in suitable areas would be the future research we will keep exploring.

Line 575:
These results suggest that effective siting of BMPs and a careful assessment of spontaneously existing (accidental) retention zones that accumulate both water and N loads from upstream can be used to achieve environmental goals for developed watersheds, by leveraging naturally occurring and built features providing ecosystem services.

o Line 481: The conclusion is very general. It needs to refer to your specific results presented before. Please elaborate whether the framework is applicable for other watersheds.

Thanks for your suggestion. We elaborated our Conclusion with referring to our results of simulated $NO_3^-$ concentration. We also specified our model can be applied to other suburban watersheds relying mainly on septic systems. Please refer to our response to your General Comment #2: Too general conclusion.

---

## Referee Report (RR1)

**hess-2023-256**

*General Comments*

The revised version of the manuscript entitled "Simulation of spatially distributed sources, transport, and transformation of nitrogen from fertilization and septic system in a suburban watershed" presents an augmented version of the RHESSys Model for simulating the impact of spatially distributed anthropogenic N and water inputs in suburban watersheds. The authors chose a well-monitored watershed to calibrate the augmented ecohydrologic model for subsurface hydraulic parameters and compare the model results against observed $NO_3^-$ concentrations in streams. The augmented modules of the model rely on a rich composition of data sets obtained in previous surveys and studies.

The results prove that the presented approach successfully improves the prediction of nutrient loads in streams by integrating spatially explicit anthropogenic N inputs. It delivers valuable insights about the impact of N retention hot spots and could help planners decide on effective sizing and situating of best management practices.

As the presented framework is transferable to simulate the ecohydrology of other watersheds with scarcer or absent nutrient load data, it is a valuable contribution to watershed restoration efforts.

I suggest accepting the manuscript for publication with minor revisions.

*Specific comments*

**1) The overall quality of the manuscript and presented Figures and results improved significantly. The current state of knowledge and the novelty of the presented approach have been well elaborated. The methodology is now presented clear and well sorted, highlighting the rich input of collected data that backs up the authors augmentation scenarios.**

**2) The length of several sentences (some are 3 to 4 lines long) should be revised.**

**3) The conclusion should be revised for redundant points.**

**4) Typos should be improved (see technical corrections in the report)**

**5) The contribution of one author is not described in the respective section. Please make sure all authors contributions are described in concordance with the CRediT contributor roles taxonomy (suggested by HESS)**

*Technical corrections*

l. 28 space/time ?

l. 29 With?

l. 69 typo: requires

l. 85 please improve the grammar

l. 91 typo soils

l. 120 add the: […] facilitates [the] scientific assessment […]

l. 223 might "we noted" at this line be a remnant of the reply to the review?

l. 224 Please correct the typo: from. Lastly

The standard deviations in the text are not showing the +- symbol right (e.g. l 341, 343 etc.)

l. 350 the unit mm must should in front of parenthesis

l. 379 please correct typo close-t- zero

l. 380 simulated instead of simulation

l. 409 why two values? "(-0.1, -0.3%) "

l. 431: please improve this sentence: Septic drainage patches (i.e., scenario septic only) was almost 5-fold higher (+368%) than the reference scenario none.

l. 211 Caption Table 1. what does physics of the parameters refer to?

Caption Figure 3: making 2 sentences will help the reader following on the complexity of the figure.

l. 371 Caption figure 4 is instead of was

l. 339: replace "turned on" with include

l. 465 Table 4 caption: I suggest for clarity to replace "others" with "each scenario"

l. 482 typo: transform

l. 494: it would be interesting to read here for how many years you suggest to spin up the model instead

l. 510: I like that you call it N retention hot spots here in the title, I suggest to consider to use this term throughout the text to differentiate more from N input hot spots which are described, too (Sec 2.4). This can increase clarity for the reader.

l. 545 missing "of"

l. 573-576 this last sentence is very long and redundant in some parts of it content

l. 578-581 Please divide this first very long sentence into several

I suggest to improve the conclusion. There are some redundant points and very long sentences covering 3 to 4 rows.

Figure A2: what values does the legend in panel a for soil texture represent?

l. 620: Figure A4 why (meter) and not (m)?

l. 645: The contribution of one author is not described. Please make sure all authors contributions are described in concordance with the CRediT contributor roles taxonomy

---

## Author Response (AR4)

**Respond Letter**

Dear editor,

We appreciate your time to handle our manuscript.  We were able to improve the quality of our manuscript with constructive comments from you and all referees.  In this round of revision, we addressed all major and minor comments from all three referees.  We also did some minor edits to clarify some concepts and terms, which we think significantly improved the readability and quality of this study.  However, we found Referee #3 (report #2) was reviewing the initial submission, not the revised manuscript.  Many of the referee's comments were addressed in the previous revised manuscript, and we carefully respond to these comments by citing the revisions we did in the previous and this round.  We wanted to let you know this and would be happy to address any other comments of Referee #3 if she/he has additional questions to our newest version of manuscript.

Please don't hesitate to let us know about any questions/issues you have regarding to the manuscript.  We appreciate your efforts and time spending with our manuscript again.

Best regards,
Ruoyu Zhang (on behalf of all co-authors)

**Report #1 (Referee #2)**

The revised version of the manuscript entitled "Simulation of spatially distributed sources, transport, and transformation of nitrogen from fertilization and septic system in a suburban watershed" presents an augmented version of the RHESSys Model for simulating the impact of spatially distributed anthropogenic N and water inputs in suburban watersheds. The authors chose a well-monitored watershed to calibrate the augmented ecohydrologic model for subsurface hydraulic parameters and compare the model results against observed NO3- concentrations in streams. The augmented modules of the model rely on a rich composition of data sets obtained in previous surveys and studies. The results prove that the presented approach successfully improves the prediction of nutrient loads in streams by integrating spatially explicit anthropogenic N inputs. It delivers valuable insights about the impact of N retention hot spots and could help planners decide on effective sizing and situating of best management practices. As the presented framework is transferable to simulate the ecohydrology of other watersheds with scarcer or absent nutrient load data, it is a valuable contribution to watershed restoration efforts.

I suggest accepting the manuscript for publication with minor revisions.

Specific comments
**1) The overall quality of the manuscript and presented Figures and results improved significantly. The current state of knowledge and the novelty of the presented approach have been well elaborated. The methodology is now presented clear and well sorted, highlighting the rich input of collected data that backs up the authors augmentation scenarios.**
**2) The length of several sentences (some are 3 to 4 lines long) should be revised.**
**3) The conclusion should be revised for redundant points.**
**4) Typos should be improved (see technical corrections in the report) #5) The contribution of one author is not described in the respective section. Please make sure all authors contributions are described in concordance with the CRediT contributor roles taxonomy (suggested by HESS)**

> Response:
>
> We appreciate the referee's thorough comments to our revised manuscript. It was your constructive suggestions that help to improve our manuscript significantly. We have carefully addressed all comments below. For your reference, all our responses are in blue color. We also quote and shade texts in the manuscript, and use red color to highlight new changes we made. We also carefully revised the entire manuscript to ensure that there is no grammar mistake found.

l. 28 space/time ?
> Thanks for the comment. We have removed "space/time" in the sentence.

l. 29 With?

> Thanks for the comment. We rephrased the sentence as:
> "With **calibrating** subsurface hydraulic parameters only and without calibrating ecosystem and biogeochemical processes, the model … "

l. 69 typo: requires
> Thanks for the comment. We corrected this as:
> "…, decision makers are facing environmental challenges which **require** detailed planning for siting BMPs effectively in watersheds …"

l. 85 please improve the grammar
> Thanks for the comment. We rephrased the sentence as:
> "However, these models lack hillslope water and nutrient mixing […]. **These interactions** are important to simulate the formation of biogeochemical hot spots **where** potential uptake and retention **of nutrients are high.**"

l. 91 typo soils
> Thanks for the comment. We correct "soils" to "**soil**".

l. 120 add the: […] facilitates [the] scientific assessment […]
> Thanks for the comment. We added "**the**" into the sentence.

l. 223 might "we noted" at this line be a remnant of the reply to the review?
> Thanks for the comment. We wanted to highlight the fact that we did not calibrate N-related parameters in this study. We changed to "we **emphasized** that […]" at line 223.

l. 224 Please correct the typo: from.
> Thanks for the comment. We removed "from" in the sentence.

Lastly The standard deviations in the text are not showing the +- symbol right (e.g. l 341, 343 etc.)
> Thanks for the comment. The $\pm$ symbol was not shown correctly. We have fixed the issue at line 348.

l. 350 the unit mm must should in front of parenthesis
> Thanks for the comment. We have moved the unit, including in other sentences throughout the manuscript, in front of the parenthesis.

l. 379 please correct typo close-t- zero
> Thanks for the comment. We fixed this as "**close-to-zero**".

l. 380 simulated instead of simulation
> Thanks for the comment. We changed to "**simulated**".

l. 409 why two values?  "(-0.1, -0.3%) "

Thanks for the comment. We added the unit, mm month$^{-1}$, to avoid the confusion of the bias and percentage of change. We also fixed this issue in the rest of the manuscript.

Line 412:
"ET in lawn patches and septic drainage fields increased to (by) 42.3 (+0.4 mm month$^{-1}$, 1.0%) and 40.8 (+6.5 mm month$^{-1}$, 18.9%) mm month$^{-}$."

l. 431: please improve this sentence: Septic drainage patches (i.e., scenario septic only) was almost 5fold higher (+368%) than the reference scenario none.

Thanks for the comment. We rephrased the sentence in Line 437 as:
"Denitrification rates in septic drainage patches was increased by 368% in scenarios *septic only* compared to in the reference scenario *none* where these patches do not receive additional water and N inputs."

l. 211 Caption Table 1. what does physics of the parameters refer to?

Thanks for the comment. We further clarified this as "physical representations" of calibrated RHESSys parameters in Line 211.

Caption Figure 3: making 2 sentences will help the reader following on the complexity of the figure.

Thanks for the suggestion. We rephrased the caption as:
"The ensemble mean of daily streamflow from simulations (red), USGS observations (blue), and the daily 95% uncertainty range (grey) from 50 simulations with NSE greater than 0.5. The periods of (a) calibration was from Oct. 2012 to Sep. 2015 and (b) validation from Oct. 2015 to Sep. 2017. All simulations include irrigation, lawn fertilization, and septic processes"

l. 371 Caption figure 4 is instead of was

Thanks for the comment. We changed the "was" to "is".

l. 339: replace "turned on" with include

Thanks for the comment. We changed "turned on" to "include".

l. 465 Table 4 caption: I suggest for clarity to replace "others" with "each scenario"

Thanks for the comment. We rephrased as:
"Absolute and relative changes between scenario none and other scenarios are included below denitrification rates."

l. 482 typo: transform

Thanks for the comment. We changed to "transformation" here.

l. 494: it would be interesting to read here for how many years you suggest to spin up the model instead

> Thanks for the comment. We mentioned that, in a previous RHESSys study (Lin et al., 2015), the spin up period was set to 500 years. However, the model was for a fully forested watershed where N input rate is much lower than our suburban watershed. With greater N input and a shorter residence time of N, we think spinning up model for 30 years resulting in stable C:N ratio, though longer spin up period could have more accurate results.
>
> Line 498:
> "A longer spin up period (i.e., 500 years) was used in Lin et al. (2015) for a fully forested watershed. In our suburban watershed with larger inputs and shorter residence time of N, the spin up period could be shorter than a fully forested watershed, as evidenced by asymptotic C:N ratio after 30 years."

l. 510: I like that you call it N retention hot spots here in the title, I suggest to consider to use this term throughout the text to differentiate more from N input hot spots which are described, too (Sec 2.4). This can increase clarity for the reader.

> Thanks for the suggestion. We thoroughly checked the manuscript, and denoted "N input" and "N retention" in front of "hot spots" to further differentiate different types of hot spots. Here are some changes we made:
>
> Line 33:
> "The highest predicted denitrification rates, or N retention hot spots, were downslope of lawn and septic locations in a constructed wetland [...]"
>
> Line 266:
> "[...] areas receive additional water and N input from septic effluents and may become N input hot spots of $NO_3^-$ in the watershed."

l. 545 missing "of"

> Thanks for the comment. We added "of" in the sentence.

l. 573-576 this last sentence is very long and redundant in some parts of it content

> Thanks for the comment. We revised the long sentence in line 579 as:
> "Our results showed the spatial pattern of N retention and identified spontaneously existing (accidental) retention zones that accumulate both water and N loads from upstream. By effective siting of BMPs based on our results for developed watersheds, both naturally occurring and built features could become N retention hot spots and provide ecosystem services to improve water quality in the future."

l. 578-581 Please divide this first very long sentence into several I suggest to improve the conclusion. There are some redundant points and very long sentences covering 3 to 4 rows.

Thanks for the suggestion. We rephrased this long sentence in line 584 as:
"Our analysis provides important insights into how different sources of N input interact with ecohydrological processes to control N export from suburban and exurban watersheds relying on local groundwater for domestic use and septic systems for wastewater release. With single-family houses dominant in these watersheds, the input of lawn fertilization and irrigation water, and septic effluent volume and N load are concentrated in limited areas at much higher per unit area rates."

Figure A2: what values does the legend in panel a for soil texture represent?

Thanks for the suggestion. We labeled the soil texture in the figure, instead of using the index for soil type, to avoid the confusion.

[Figure]

l. 620: Figure A4 why (meter) and not (m)?

Thanks for the suggestion. We have changed the meter to "m" in the figure.

[Figure]

l. 645: The contribution of one author is not described. Please make sure all authors contributions are described in concordance with the CRediT contributor roles taxonomy

Thanks for the comment. We included the contribution of Dr. Laurence Lin here for helping develop and calibrate the RHESSys model in line 661.

"LL provided technical assist on developing and calibrating the RHESSys model, and PMG, AKS, JMD, and AJG provided water chemistry and biogeochemical data. All authors reviewed and edited the paper."

**Report #2 (Referee #3)**

Major comment: My main concern is the lack of rigor in model calibration.

First, though the model was evaluated on water and nitrogen dynamics, the model was only learned against the streamflow observations with limited parameters. Since N is the focus of the study, the parameters used in the ferritization and the septic system (i.e., Eqs (1)-(4)) should also be estimated based on the N observations. Otherwise, while focusing on transport and transformation, only the transport part of the model was well represented, which logically does not follow through.

Second, the flow calibration might be problematic. As major components of the water cycle, the ET and irrigation processes are not calibrated. Both potentially contributed to the underestimation of the low flow period in Figure 3. Although the authors admit this caveat in the discussion, it is suggested to tune the parameters of the two processes and discuss how the underestimation might affect the conclusion if unsuccessful.

Third, there is a lack of details in the current model calibration in Section 2.2, including
- I believe the authors generated an ensemble of model simulations and picked up the best run. If so, how many runs were performed? How was the parameter ensemble generated?
- Did the authors split the observed time period into training and test periods? It is recommended that this is done to limit the adverse impact of temporal extrapolation.
- There are two gages in the catchment (Lines 145-147). Have both gages been used to estimate model parameters? Or only the USGS one?

> ### *Response:*
>
> Dear reviewer, we appreciate your time and suggestions to improve our model. Our manuscript focused on the transformation of $NO_3^-$ through denitrification and evaluated the changes of terrestrial denitrification at many locations, including hot spots at riparian areas and wetlands (Figure 5 and Table 4). For your reference, all our responses are in blue color. We also quote and shade texts in the manuscript, and use red color to highlight new changes we made. We also carefully revised the entire manuscript to ensure that there is no grammar mistake found.
>
> For **calibration**, we substantially revised our calibration approach in our revised manuscript, compared to the initial submission which we believe was the version accidentally assigned to your review. Here are some main improvements we had in the revision:
>
> Firstly, in the revised manuscript, we included 50 behavioral simulations, which yield 1) NSE of streamflow from water year 2013 to 2015 greater than 0.5 and 2) gw2 lower than 0.5, to quantify the uncertainty of ecohydrological responses of our model (see the

uncertainty ranges in Figure 3 and 4). This improvement was based on the constructive comments from reviewers in the first round.

Secondly, there are no ET and irrigation observations that allow us to perform calibration for related parameters in this watershed. We would like to further calibrate the ET and irrigation in the future once high-resolution data are available. Otherwise, we don't want to overfit our model on streamflow data, which reduced the model's degree of freedom and increased bias potentially.

Lastly, we wanted to highlight that one significant aspect of our study is to restrict our calibration on soil and subsurface hydraulic parameters only. This allowed us to evaluate how well our model could be generalized to assess N dynamics for most ungagged suburban watersheds where only discharge, but no water chemistry (expensive to obtain) observations, are existing. We discussed this in Introduction and Discussion:

In Introduction (Line 103):
"The framework should be capable of extension to watersheds without water chemistry data which are less available than discharge records worldwide. It would be a valuable feature of the framework to estimate nutrient dynamics reasonably, while restricting calibration to hydrologic parameters. Calibrating nutrient dynamics may not allow generalization to watersheds without chemistry records or extrapolation to conditions in which water quality BMPs are implemented."

Other minor revisions:
Line 34: best BMPs --> BMPs
   Thanks for the comment. We deleted the "best" in our revised manuscript.

Figure 1: Please show the locations of POBR, the two gages, and the riparian area.
   Thanks for the comment. The pond branch (POBR) is outlined in grey, and two gages are shown as red stars in Figure 1. Including the riparian areas (defined as height above nearest drainage, HAND, lower than 1.5 m) would make Figure 1 quite busy. The riparian areas are basically identical to dark blue (saturated) areas in Figure A4.

[Figure]

**Figure 1. Study watershed Baisman Run (BARN) in suburban Baltimore County, Maryland (from ESRI). The inset highlights two N retention "hot spots": A sediment accumulation zone (upper circle) receiving drainage from roads and a constructed wetland (lower circle). Pond Branch (POBR), a fully forested subcatchment is outlined in grey.**

[Figure]

**Figure A4. Spatial pattern of ensemble mean water table depth (m) of Baisman Run during the entire study period (water year 2013 to 2017) from the 50 behavioral simulations under scenario *none*. Map in projection NAD83 UTM 18N (EPSG: 26918).**

Lines 152-153: Please present the spin-up storage of soil C and N.

Thanks for the comment. The ratio of soil C to N at the entire watershed was stabilized to 8.5 after the spin-up. We added this information at line 179:

"Inspection of the spin-up storage of soil C and N showed they were asymptotic with stable C:N ratios, with a mean of 8.5 in the entire watershed."

Section 2: Please provide a conceptual diagram and the key mathematical representation of nitrogen cycling (e.g., denitrification) used by RHESSys and link it to the proposed fertilization and septic system.

Thanks for the suggestion. The details of nitrogen cycling of RHESSys has been documented in Figure 2 of Lin et al. (2015). We mentioned in the manuscript to refer readers to check this diagram:

Line 254 & 284:
"Once $NO_3^-$ is released to soil, N cycling is simulated following the procedure detailed in Lin et al. (2005)."

"Once $NO_3^-$ is added to surface detention storage, N cycling is simulated following the procedure detailed in Lin et al. (2005)."

[Figure]

Figure 2 from Lin et al. (2005) showing detailed N cycling in RHESSys

Table 1: What are s,sv, svalt, and gw abbreviated for? Please illustrate.

Thanks for the comment. We included the physical meanings of these abbreviations in Table 1 as below:

Table 1. RHESSys parameters being calibrated and their **physical representations** (Tague and Band, 2004). Calibrated results shown as ranges of multipliers to original soil properties (Fig. A2 & A3) and groundwater component generating behavioral simulations with NSE greater than 0.5 for streamflow.

| Parameter Groups | RHESSys Parameter Abbreviations | | Detail | Source | Unit | Multiplier Range |
|---|---|---|---|---|---|---|
| Lateral soil hydraulics | s | $m_l$ | Decay rate of lateral saturated hydraulic conductivity with depth | USDA SSURGO, 2019 | - | 0.31–2.91 |
| | | $K_{sat0\_l}$ | Lateral saturated hydraulic conductivity at the soil surface | | m day$^{-1}$ | 0.38–2.93 |
| | | $z$ | Soil depth | | m | 1.65–5.95 |
| Vertical soil hydraulics | sv | $m_v$ | Decay rate of vertical saturated hydraulic conductivity with depth | | - | 0.51–1.98 |

| | | | | | | |
|---|---|---|---|---|---|---|
| | | $K_{sat0\_v}$ | Vertical saturated hydraulic conductivity at the soil surface | USDA SSURGO, 2019 | m day$^{-1}$ | 0.52–1.98 |
| Soil properties | svalt | $b$ | Pore size index | USDA SSURGO, 2019 | - | 0.51–1.98 |
| | | $\varphi_{ae}$ | Air entry pressure | | pounds inch$^{-2}$ | 0.5–1.05 |
| Groundwater dynamics | gw | $gw_1$ | Fraction of bypass from the saturated zone to groundwater storage | | - | 0–0.13 |
| | | $gw_2$ | Fraction of loss from groundwater storage to stream | | - | 0.03–0.5 |
| | | $gw_3$ | Fraction loss from surface to groundwater storage | | - | 0–0.07 |

Lines 190-191: The authors state that "Both surveys were conducted during significant drought conditions (2002 and 2008) when lawncare was reduced due to groundwater supply concerns". Should irrigation be increased to account for drought conditions?

> Thanks for the comment. Since there is no water supply sewer serving the watershed, groundwater is the only source for domestic water use. During severe drought, Maryland government may enforce water use restrictions (https://mde.maryland.gov/programs/water/droughtinformation/pages/restrictions.aspx), and homeowners may use water for prioritized purposes (e.g., sanitary and drinking) rather than irrigating lawns.

Eq.(2): The authors set a limit to the available irrigation water based on GW storage in the below cell. Should it be represented by a pipe-based irrigation system that extracts water from the entire GW pool instead of the below GW? That would change the result of water distribution, though.

> Thanks for the comment. In our study watershed, water extraction was implemented at individual households. No center or piped irrigation system (or external water supply) exists in this suburban watershed. Also, RHESSys assumes a uniform groundwater storage in each hillslope (total of 16 in BARN, Figure A7), and irrigation water is extracted from there if one identified well is located within it. Lastly, as mentioned above, this irrigation module requires further data and study to be improved to better capture the practices at household level.

[Figure]

**Figure A7. Hillslope indices of Baisman Run. Map in projection NAD83 UTM 18N (EPSG: 26918).**

Eqs.(1) and (4): It is unclear how the authors set the parameters LF and IRmax. See my main comment above.

> Thanks for the comment. Since Leaching Fraction (LF) could be confusing to readers, we changed it to release fraction (RF) in equation 1. Our model estimates RF by assuming 1) an exponential decay of fertilizer (similar to commonly found slow-release fertilizer) and 2) 10% of input fertilizer would remain after one fertilization interval (FI). This assumption could be overwritten if users have observations of fertilization practices.

> Line 247:
> "Assuming all lawn fertilization is done with the slow-release fertilizer designed to remain 10% after one fertilization interval, the daily release fraction ($RF$) is determined by the fertilization interval ($FI$), following Eq. (1):
> $$RF = -\frac{\log 0.1}{FI}, \tag{1}$$
> In our case study, our 60-day fertilization interval results in 3.8% of nutrients in the fertilization pool to decline exponentially and transported to other pools per day and then stored, consumed by vegetation, immobilized, denitrified or further transported to groundwater and downslope. User-defined fertilization time series could overwrite this setting of lawn fertilization if observations are available."

> The maximum daily **irrigation** was based on EPA's recommendation of about 1 inch per week (or ~4 mm per day) of irrigation.
> We added the reference in our manuscript at line 301 as:

> "In the current model, we defined the maximum irrigation rate ($IR_{max}$) in BARN as 4 mm day$^{-1}$, which was converted based on the EPA's recommendation (U.S. EPA, 2024) of

one inch per week for lawns. This rate can be modified based on the local practices or for sensitivity analysis."

Lines 282-283 and Figure 3: The underestimation of low flow is during irrigation season. So, it could be attributed to either irrigation or ET, or both. A rigorous calibration of the two processes is needed. See my main comment above.

Thanks for the comment. As we mentioned above, there are no ET or irrigation dataset available in the watershed to calibrate our model. Calibrating additional parameters could introduce overfitting and reduce the degree of freedom (ability of generalization) of our model. We would like to improve the model's performance on ET, irrigation, and streamflow in the future once we have high-quality data in our study watershed.

Lines 291-292: How was the "mean streamflow-weighted long-term observed concentration" calculated? Please provide the equation.

Thanks for the comment. In our revised manuscript, we no longer use the streamflow-weighted concentration, but the absolute concentration (Figure 4).

[Figure]

Figure 4. Ensemble weekly mean of (a) $NO_3^-$ concentration and (b) load at the outlet of Baisman Run over the entire study period (water year 2013 to 2017). The 95% uncertainty boundary for scenario *both* is shown in grey.

Figure 5: Both nitrate load and concentration were underestimated for all cases. Again, the transformation is not calibrated.

Thanks for the comment. We apologize for showing the incorrect plot in our initial submission. Please refer to the new Figure 4 (shown above) to see how our model performed on simulating $NO_3^-$ dynamics.

Section 3.3.2: Please provide the conceptual diagram of denitrification and its linkage to fertilization and the septic system.

Thanks for the comment. Please refer to the conceptual diagram, shown in our respond to the comment for "Section 2".

Figure 6: The difference (right row) is difficult to tell. Please either reduce the color bar range or use a different colormap to better illustrate the difference.

Thanks for the comment. We revised the figure as below:

[Figure]

**Figure 5. Ensemble mean differences of water table depth (top panel) and denitrification (lower panel) between scenario *none* and scenario *fertilizer only* (a & d), *septic only* (b & e), and *both* (c & f). The two hot spots of denitrification (i.e., wetlands in Fig. 1) were circled in (f).**

Line 369: "… were in spring when fertilizer is applied to lawns and soil moisture is generally higher" --> "… were in spring when fertilizer was applied to lawns and soil moisture was generally higher"

Thanks for the suggestion. We used present tense here to show that the time of fertilization is generally in spring when soil is commonly wet. Using past tense could introduce the confusion that fertilization now is no longer initially applied in spring.

Line 370: "… had significant increases in denitrification in winter when the watershed receives"
--> "… had significant increased in denitrification in winter when the watershed received"

> Thanks for the suggestion. We used present tense here to show that riparian areas generally receive more $NO_3^-$ during winter, due to stronger subsurface flow (no ET) and low N retention rate (low denitrification).

Sections 4.1 and 4.2: The authors discussed the underestimation of both low flow and nitrate in multiple places and attributed it to uncertainty in ET/nitrogen processes without calibrating them. Please see my main comment.

> Thanks for the comment. As mentioned above, we wanted to keep the calibration simple (only on limited parameters) considering ET and water chemistry data are much less available than discharge for most gaged watersheds. Calibrating additional parameters could introduce overfitting and reduce the degree of freedom (ability of generalization) of our model.
>
> In addition, we also wanted to evaluate whether our model can reasonably capture N dynamics assuming we only have discharge data that could be used for calibration. Our results of $NO_3^-$ concentration and denitrification showed the reliable capacity of our model to be applied to vast watersheds without water chemistry data.

**Report #3 (Referee #4)**

Overall Comments:

> *Response:*
>
> We appreciate the referee's thorough comments to our revised manuscript. It was your constructive suggestions that help to improve our manuscript significantly. We have carefully addressed all comments below. For your reference, all our responses are in blue color. We also quote and shade texts in the manuscript, and use red color to highlight new changes we made. We also carefully revised the entire manuscript to ensure that there is no grammar mistake found.

1) Please clearly state, if accurate, that the simultaneous consideration of fertilization, septic systems, and irrigation schemes in the RHESSys model is novel to this study. Emphasize this point in both the abstract and introduction.

> Thanks for the suggestion. Our augmented RHESSys model developed new modules of fertilization, septic, and irrigation processes at household/parcel level. Other RHESSys versions allow users to provide time series of irrigation and fertilization, but not at household level. Our model allows to simulate these processes more realistically for suburban watersheds like Baisman Run. We emphasized these in both Abstract and Introduction:
>
> In Abstract (line 27):
> "We augmented a distributed ecohydrological model, RHESSys, with estimates of the spatial distribution of these loads at household parcel level to develop a predictive understanding of the factors generating upland and riparian nitrogen cycling, transport and stream $NO_3^-$ concentrations."
>
> In Introduction (line 112):
> "In this study, we augmented RHESSys to include household-level transfer of groundwater for lawn irrigation [...]. [...]. In summary, by adding modules of household-level lawn irrigation, fertilization, and septic releases (see Sect. 2.3) that are commonly found in suburban areas, RHESSys is designed [...]."

2) Regarding the irrigation scheme, I am curious about its accuracy. Typically, farmers do not irrigate based solely on PET/ET conditions but rather follow conventional practices. Additionally, some areas may require more or less irrigation for various reasons. It would be beneficial if the authors could demonstrate the feasibility of this irrigation scheme by comparing it with satellite-based irrigation detection methods (though this is not mandatory) or by citing reliable sources. This is especially important given that the irrigation scheme is applied throughout the study period.

Thanks for the suggestion. Our current method (using ET/PET) is approximating the irrigation demand while considering the spatial distribution of lawns and households in watersheds. We discussed that the method cannot consider the actual heterogeneity of irrigation practices of people, including irrigation amounts and timing:

"For irrigation, our model applies irrigation close to its maximum (4 mm day$^{-1}$) when water stress is high, but residents may not irrigate their lawns at these rates during drought to conserve groundwater, and may continue to irrigate lawns during wet periods with automated sprinkler systems."

Though we would like to better capture the irrigation signals at household level, there is no dataset available for irrigation at the resolution (< 100 m) we needed to account for the heterogeneity of irrigation practices in our watershed (just 3.8 km$^2$). In other words, we need more survey and high-resolution observations in the future to account for these household-level heterogeneity of irrigation. We added a sentence in the Discussion in line 543 to highlight this limitation:

"Surveys and high-resolution satellite observations could help to improve our irrigation module and accurately estimate the timing and quantity of irrigation practices in suburban watersheds."

Specific Comments:

Line 33: Remove "in the range of measured values" and include the exact numbers.

Thanks for the suggestion. The range of measured denitrification rates from Suchy et al. (2023) and Raciti et al. (2011) varies a lot. We would need more information to help readers understand these, and this would make our abstract too long. We think it would be better for reader to check the Section 3.3.2 for the details of denitrification.

Line 50: Did you consider spontaneously developed "hot spots" in the model in this study?

Thanks for the comment. We discussed the denitrification rates in the constructed and accidental wetlands (Fig. 1) were highest within the watershed at line 441:

"The annual denitrification rates in the sedimentation accumulation zone (upper red circle in Fig. 5) showed a significant increase after activating fertilization and septic processes, from 76.9 kg N ha$^{-1}$ year$^{-1}$ before to 95.6 (+18.7, 24.3%) kg N ha$^{-1}$ year$^{-1}$ after activation. Similarly, denitrification rates in the constructed wetland (lower red circle in Fig. 1) increased from 81.5 kg N ha$^{-1}$ year$^{-1}$ before to 102.7 (+21.2, 26%) kg N ha$^{-1}$ year$^{-1}$ after activation."

And we further discussed these locations as hot spots in the watershed at line 529:
"Specifically, these two wetlands covering only 0.09% of the watershed contributed to 0.39% of the total denitrification during the study period."

Line 108: Define "RHESSys" earlier in the manuscript, as it appears before Line 108.

Thanks for the comment. We moved the full name of RHESSys to line 89 as:
"Fully distributed hydrology models, such as […] RHESSys (Regional Hydro-Ecological Simulator System, Tague & Band, 2004) could explicitly […]"

Line 134: BARN has already been defined; there is no need to define it again.

Thanks for the suggestion. We revised the sentence in line 134 as:
"Our study watershed (Fig. 1), BARN, is in Baltimore County, MD, […]"

Line 145: Is the temperature range accurate? Is there no snowfall?

Thanks for the comment. Baltimore has a moderate and wet climate. The mean annual maximum and minimum temperatures are as reported. Snow does occur in Baltimore in the wintertime.

Line 155: Are there other facilities, such as tennis courts or soccer fields, in this area that could alter the land properties?

Thanks for the comment. Except for forest, sparse single-family houses are the dominant land use in BARN. No large facilities, such as schools or sport fields, are found in the watershed by inspecting Google Earth satellite imageries.

Line 179: Please specify the ratio mentioned.

Thanks for the comment. The ratio of soil C to N at the entire watershed was stabilized to 8.5 after the spin-up. We added this information at line 180:

"Inspection of the spin-up storage of soil C and N showed they were asymptotic with stable C:N ratios, with a mean of 8.5 in the entire watershed."

Line 205: Define SSURGO before using the term.

Thanks for the comment. We added the full name of SSURGO here as:
"[…] initial estimates (Fig. A2) from the Soil Survey Geographic Database (SSURGO, USDA, 2019)"

Line 218: Please specify these limits in the supplementary information or provide references.

Thanks for the comment. We added the reference, Smith et al. (2022), here in line 206.

"From four thousands of parameter set realizations randomly chosen within specified limits described in Smith et al., 2022, behavioral sets are chosen as yielding Nash-Sutcliffe efficiency […]"

Line 255: Provide a reference to confirm that all households in BARN use septic systems. Do they all use the same system? How do you ensure that the septic systems were functioning properly during the study period?

Thanks for the comment. BARN is not within the Baltimore's Urban Rural Demarcation Line (Zoning regulations of Baltimore, https://perma.cc/LCC2-8BZJ), outside which no centralized sewer systems, including water supply and sanitary sewage, exist. Therefore, all households use their own septic systems for wastewater drainage.

As each household heavily relies on the septic system for daily life, we believe homeowners would quickly fix issues if septic systems were not functioning. As the result, our study did not consider these incidents, assuming the impacts of short infunctioning are negligible.

Line 297: Explain why 4 mm/day was chosen as the maximum irrigation rate in BARN. Any references?

Thanks for the comment. The maximum daily irrigation was based on EPA's recommendation of about 1 inch per week (or ~4 mm per day) of irrigation.
We added the reference in our manuscript at line 301 as:

"In the current model, we defined the maximum irrigation rate ($IR_{max}$) in BARN as 4 mm day$^{-1}$, which was converted based on the EPA's recommendation (U.S. EPA, 2024) of one inch per week for lawns. This rate can be modified based on the local practices or for sensitivity analysis."

Line 341: ?

Thanks for the comment. The $\pm$ symbol was not shown correctly. We have fixed these syntax issues.

Line 350: Provide an explanation for the underestimation of streamflow from July to October 2016. Additionally, please include the time series of WSF values in the supplementary information.

Thanks for the comment. For the streamflow underestimation, we discussed at line 467 as:

"We also noted that our model tended to underestimate the lowest streamflows during the growing season, which was also found in another suburban watershed, Dead Run, in Baltimore by Miles (2014). Several potential reasons could cause this discrepancy: 1) Higher transpiration estimates caused by uncertainties in vegetation ecophysiological parameters in RHESSys controlling vegetation water use or phenology; 2) Underestimation of groundwater recharge and release to streams during the growing season; and 3) A lack of household modulation of groundwater use during dry periods. During our prior surveys (Law et al., 2004; Fraser et al., 2013) residents stated they had reduced their water use during droughts. While the model underestimation was negligible, additional empirical data about water flux, groundwater processes, and

household water management would enhance model prediction accuracy of
hydrological processes, especially during the growing season."

We included the watershed-scale water stress factor at each patch (Eq. 3) as Figure A6:

[Figure]

**Figure A6. Spatial pattern of ensemble mean water stress factor (ET/PET) of Baisman Run during the entire study period (water year 2013 to 2017) from the 50 behavioral simulations under scenario *none*. Map in projection NAD83 UTM 18N (EPSG: 26918).**

Figure 5: Consider adding arrows to highlight the circles, as they are difficult to see.

Thanks for the comment. We added an inset to highlight the two wetlands (circles).

[Figure]

**Figure 5. Ensemble mean differences of water table depth (top panel) and denitrification (lower panel) between scenario *none* and scenario *fertilizer only* (a & d), *septic only* (b & e), and *both* (c & f). The inset highlights two hot spots of denitrification (i.e., wetlands in Fig. 1) were circled in (f).**